# GriddlyJS: A Web IDE for Reinforcement Learning

**Christopher Bamford**[*]
Queen Mary University
c.d.j.bamford@qmul.ac.uk

**Minqi Jiang**
Meta AI & UCL
msj@meta.com

**Mikayel Samvelyan**
Meta AI & UCL
samvelyan@meta.com

**Tim Rocktäschel**
UCL
tim.rocktaschel@ucl.ac.uk

## Abstract

Progress in reinforcement learning (RL) research is often driven by the design of new, challenging environments—a costly undertaking requiring skills orthogonal to that of a typical machine learning researcher. The complexity of environment development has only increased with the rise of procedural-content generation (PCG) as the prevailing paradigm for producing varied environments capable of testing the robustness and generalization of RL agents. Moreover, existing environments often require complex build processes, making reproducing results difficult. To address these issues, we introduce GriddlyJS, a web-based Integrated Development Environment (IDE) based on the Griddly engine. GriddlyJS allows researchers to visually design and debug arbitrary, complex PCG grid-world environments using a convenient graphical interface, as well as visualize, evaluate, and record the performance of trained agent models. By connecting the RL workflow to the advanced functionality enabled by modern web standards, GriddlyJS allows publishing interactive agent-environment demos that reproduce experimental results directly to the web. To demonstrate the versatility of GriddlyJS, we use it to quickly develop a complex compositional puzzle-solving environment alongside arbitrary human-designed environment configurations and their solutions for use in automatic curriculum learning and offline RL. The GriddlyJS IDE is open source and freely available at `https://griddly.ai`.

## 1 Introduction

Deep reinforcement learning (RL) has seen rapid progress over the past decade, with recent methods producing policies that match or exceed human experts in tasks ranging from games like Go and Chess [52, 53] to advanced scientific applications such as plasma control [14] and chip design [37]. Concurrent with this progress is the proliferation of increasingly challenging RL environments that serve as the necessary experimental substrates for such research breakthroughs [12, 35]. In RL, environments testing specific agent capabilities serve as crucial measuring sticks against which progress is assessed, playing an analogous role to benchmarks such as MNIST [16], ImageNet [15], and GLUE [61] in supervised learning. For example, the Procgen Benchmark tests for generalization [12], and D4RL, for offline RL performance [21]. Developing RL methods for new problem settings requires new environments specifically embodying such settings. Environment development thus plays a crucial role in research progress. Moreover, the availability of a large, diverse selection of environments mitigates the risk of overfitting our methods to a small set of tasks [12, 35].

---

[*]Work done at Meta AI.

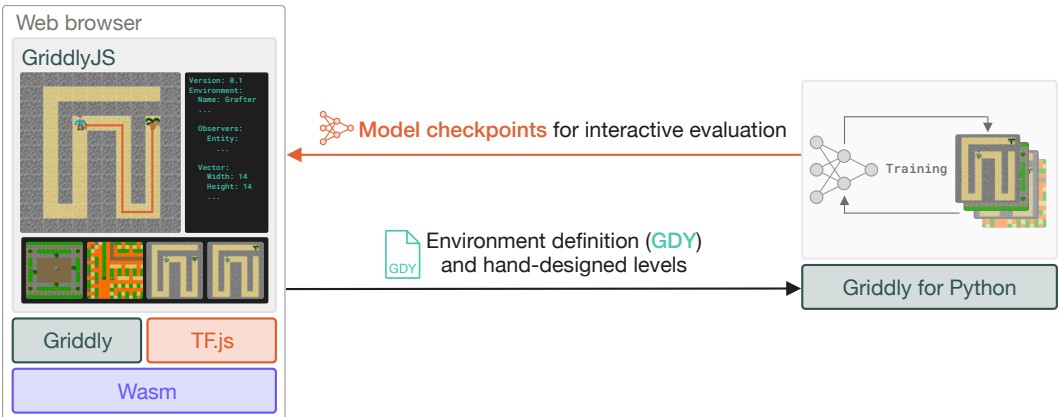

Figure 1: An overview of the human-in-the-loop environment development workflow enabled by GriddlyJS, built on top of the Griddly engine in Wasm and Tensorflow.js (TF.js). Environments and custom designed levels can be loaded into Griddly for Python for training, and model checkpoints can be directly loaded into GriddlyJS for visual evaluation.

Unfortunately, developing new environments that provide the necessary challenges for RL methods is a costly process, requiring deep expertise in software engineering, high performance computing (for efficient distributed training), and game design. This skillset is closer to that of a videogame developer than a typical machine learning researcher. Recently, procedural content generation (PCG) has emerged as the standard paradigm for developing environments that can vary throughout training, enabling the study of systematic generalization and robustness in RL [47, 35, 12, 11, 33, 49]. The more complex programming logic entailed by PCG algorithms, i.e. creating probabilistic programs that specify distributions over environment configurations, adds considerable engineering overhead to the creation of new RL environments [4]. As environments grow in number and complexity, researchers pay the additional cost in managing their associated build pipelines and dependencies—an often time-consuming task. This overhead can make reproducing results a difficult ordeal, even when pre-trained model checkpoints are provided. Moreover, most environments do not offer any tooling for visualizing, evaluating, or recording agent trajectories. Currently, writing code to support these activities demands a large time commitment from researchers, slowing down the pace of research progress and adding further obstacles to debugging and assessing existing methods.

To address these challenges, we introduce GriddlyJS, a web-based integrated development environment (IDE) based on a WebAssembly (Wasm) version of the Griddly engine [6]. GriddlyJS provides a simple interface for developing and testing arbitrary, procedurally-generated grid-world environments in Griddly using a visual editor and domain-specific language based on YAML, with support for highly complex game mechanics and environment generation logic. The visual editor allows rapid design of new levels (i.e. variations of an environment) via a simple point-and-click interface. Environments can be tested via interactive control of the agent directly inside the IDE.

GriddlyJS produces Griddly game description YAML files (GDY), which define environments and custom levels. GDY files can be loaded directly into Griddly for Python, producing a Gym-compatible environment. In addition, any agent model can be loaded into the GriddlyJS IDE, once easily converted to the TensorFlow.js [TF.js; 55] format, allowing visualizing, evaluating, or recording performance. The integrated development and visualization provided by GriddlyJS enables a whole new mode of closed-loop development of RL environments, in which the researcher can rapidly iterate on environment design based on the behavior of the agent. This allows designing environments that specifically break state-of-the-art RL methods, thereby more quickly pushing the field forward.

Importantly, GriddlyJS runs the Griddly runtime directly inside the browser. The environment simulation is rendered as a modular web component based on the React library [2]. This design allows researchers to easily publish fully interactive demos of pre-trained agent models in Griddly environments directly on the web, as embedded components in a webpage. This simplified sharing of agent-environment demos allows new RL results to be rapidly and comprehensively verified. It also enables the collection of novel, challenging environments designed by humans, providing similar benefits for improving robustness as recent adversarial human-in-the-loop training strategies in the supervised question and answering (QA) domain [34]—all without building and installing any additional software.

In the remainder of this paper, we provide a detailed description of the GriddlyJS IDE and highlight how it addresses existing environment-centric bottlenecks to RL research. As an exemplary use-case, we then demonstrate how GriddlyJS can be used to quickly design a new environment that require solving complex, compositional puzzles, as well as efficiently generate a new offline RL dataset based on human play-throughs in this environment. Additionally, we show how GriddlyJS can be used to rapidly produce human-designed levels, which we demonstrate empirically to be a promising approach for producing more difficult levels for RL agents than procedural content generation.

## 2 Background

### 2.1 Reinforcement Learning

GriddlyJS streamlines the design of reinforcement learning (RL) environments. Such environments can generally be represented as a partially-observable Markov Decision Process [POMDP; 56], defined as a tuple $\mathcal{M} = (S, A, O, \Omega, \mathcal{T}, R, \gamma, \rho)$, where $S$ is the state space, $A$ is the action space, $O$ is the observation space, $\Omega : S \rightarrow O$ is the observation (or emission) function, $\mathcal{T} : S \times A \rightarrow S$ is the transition function, $R : S \rightarrow \mathbb{R}$ is the reward function, $\gamma$ is the discount factor, and $\rho$ is the distribution over initial states. At each time $t$, the RL agent takes an action $a_t$ according to its policy $\pi(a_t|o_t)$, where $o_t \sim \Omega(s_t)$, and the environment then transitions its state to $s_{t+1} \sim \mathcal{T}(s_t, a_t)$, producing a reward $r_t = R(s_{t+1})$ for the agent. The RL problem seeks to maximize the expected return, that is, the sum of future discounted rewards over the initial state distribution $\rho$. Given the policy $\pi$ is parameterized by $\theta$, this goal is captured by the objective $J(\theta) = \mathbb{E}_\pi \left[ \sum_t \gamma^t r_t \right]$, where the expectation is over state-action trajectories arising from following $\pi$ in $\mathcal{M}$ with $s_0 \sim \rho$.

### 2.2 Procedural Content Generation

Recent focus in RL research has shifted from singleton environments, which take exactly the same form across each training episode, to environments making use of procedural content generation (PCG) that can vary specific aspects throughout training. Policies trained on singleton environments, like those in the Atari Learning Environment Benchmark [8] are overfit to the single environment instance, failing when even small changes are applied to the original training environment [20]. In contrast, PCG environments produce an endless series of environment configurations, also called *levels*, by modifying specific environment aspects algorithmically per episode, e.g. the appearance, layout, or even specific transition dynamics. PCG environments can typically be deterministically reset to specific levels by conditioning the underlying PCG algorithm on a random seed.

PCG environments allow us to evaluate the robustness and systematic generalization of RL agents. The evaluation protocol follows that in supervised learning, whereby the agent is trained on a fixed number of training levels and tested on held-out levels. Environment levels typically share a common dynamical structure, including a shared state and action space, allowing for learning transferable behaviors across levels. As a result, providing the agent with more levels at training typically leads to stronger test-time performance [12]. A separate, commonly used training method called *domain randomization* [DR; 59] simply resets the environment to a random level per episode, based on the underlying PCG algorithm. DR can produce robust policies and can be viewed as a form of data augmentation over the space of environment configurations. Adaptive curricula methods that more actively generate levels have been shown to further improve robustness in many PCG domains [43, 17, 30]. Most real-world domains exhibit considerable diversity, making PCG an important paradigm for producing environments better equipped for sim2real transfer.

### 2.3 The Case for Grid Worlds

Grid worlds are environments corresponding to MDPs with discrete actions and states that can be represented as a 3D tensor. Note that while the state is constrained to be a tensor with dimensions $M \times N \times K$, where $M, N, K \in \mathbb{Z}^+$, the actual observations seen by the agent may be rendered differently, e.g. in pixels or as a partial observation. Typically, in 2D grid worlds, each position in the grid, or *tile*, corresponds to an entity, e.g. the main agent, a door, a wall, or an enemy unit. The entity type is then encoded according to a vector in $R^K$. By constraining the state and action space to simpler, discrete representations, grid worlds drastically cut down the computational cost of training

RL agents without sacrificing the ability to study the core challenges of RL, such as exploration, planning, generalization, and multi-agent interactions.

Indeed, many of the most challenging RL environments, largely unsolved by even the latest state-of-the-art methods, are PCG grid worlds. For example, Box-World [64] and RTFM [66] are difficult grid worlds that require agents to perform compositional generalization; many games of strategy requiring efficient planning such as Go, Chess, and Shogi may be formulated as grid worlds; and many popular exploration benchmarks, such as MiniHack [49] and MiniGrid [11] take the form of grid worlds. A particularly notable grid world is the NetHack Learning Environment [NLE; 35], on which symbolic bots currently still outperform state-of-the-art deep RL agents [25]. NLE pushes existing RL methods to their limits, requiring the ability to solve hard exploration tasks and strong systematic generalization, all in a partially-observable MDP with extremely long episode lengths. Crafter is a recent grid world that features an open-world environment in which the agent must learn dozens of skills to survive, and where the strongest model-based RL methods are not yet able to match human performance [24]. To demonstrate the potential of GriddlyJS, we use it to create a Griddly-based Crafter in Section 4.

Their common grid structure and discrete action space allows for grid worlds to be effectively parameterized in a generic specification. GriddlyJS takes advantage of such a specification to enable the mass production of diverse PCG grid worlds encompassing arbitrary game mechanics. Thus, while GriddlyJS is limited to grid worlds, we do not see this as significantly limiting the range of fundamental research that it can help enable. Still, it is important to acknowlege that grid worlds can not provide an appropriate environment for all RL research. In particular, grid worlds cannot directly represent MDPs featuring continuous state spaces, including many environments used in robotics research such as MuJoCo [60] and DeepMind Control Suite [58]. Nevertheless, as we previously argue, grid worlds capture the fundamental challenges of RL, making them an ideal testbed for benchmarking new algorithmic advances. Further, many application domains can directly be modeled as grid worlds or quantized as such, e.g. many spatial navigation problems, video games like NetHack [35], combinatorial optimization problems like chip design [38], and generally any MDP with discrete state and action spaces.

## 2.4 Griddly

Griddly[2] [6] is a game engine designed for the fast and flexible creation of grid-world environments for RL, with support for both single and multi-agent environments. Griddly simplifies the implementation of environments with complex game mechanics, greatly improving research productivity. It allows the underlying MDP to be defined in terms of simple declarative rules, using a domain-specific language (DSL) based on Yet Another Markup Language (YAML). This is a similar approach to GVGAI [42], MiniHack [49] and Scenic4RL [4], where a DSL language is used to define environment mechanics. Griddly's DSL is designed to be low level in terms of interactions between defined objects, but does not go as far to allow the user to define physical models unlike Scenic4RL. This is similar in regards to GVGAI and MiniHack as they are both grid-world games. Griddly's DSL contains higher level functions such as A* search and proximity sensing, which can be configured to build higher-level behaviours for NPCs. MiniHack's DSL gives access to all of the objects within the base NetHack game, and allows them to be configured at a high level using modifiers such as "hostile" or "peaceful". The choice of using YAML is also more flexible, as YAML is a common DSL with supporting libraries in many different languages. This allows the generation and manipulation of GDY files without requiring the construction of parsers or serializers. Integrations of higher level tooling such as GriddlyJS are made possible due to this. Griddly game description YAML files share a similar structure composed of three main sections: `Environment`, `Actions`, and `Objects`. `Objects` describe the entities within the environment, declare local variables and initialization procedures. `Actions` define local transition rules determining the evolution of the environment state and any rewards that are received by the agent. Such `Actions` are distinct from the MDP actions and can be thought of as local transition functions, whose arguments are game entities (and their internal states) and the actions taken by the agent. The `Environment` section defines the environment description, e.g. its name, as well as the MDP observation and action spaces. For all environments, Griddly uses the canonical action space layout from [5], with support for invalid action masking. In

---

[2]Documentation, tutorials, examples and API reference can be found on the Griddly documentation website `https://griddly.readthedocs.io`

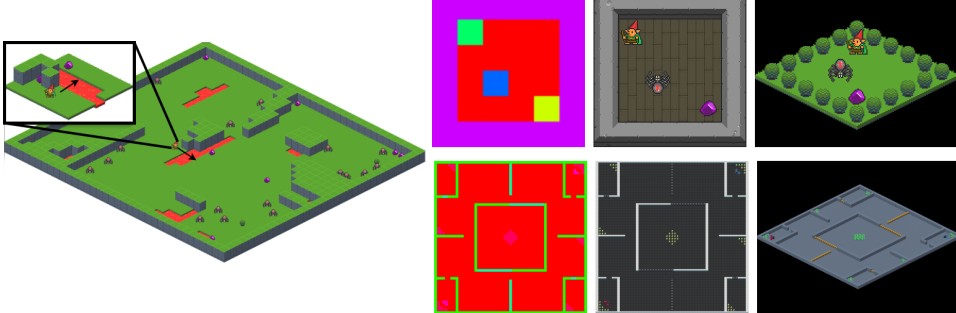

Figure 2: Visualization of some observation spaces supported in Griddly. The left figure shows an isometric, global view of an environment with an example of a local observation. On the right, each row shows the same environment state rendered under three different observers: `Vector`, `Sprite2D`, and `Isometric`.

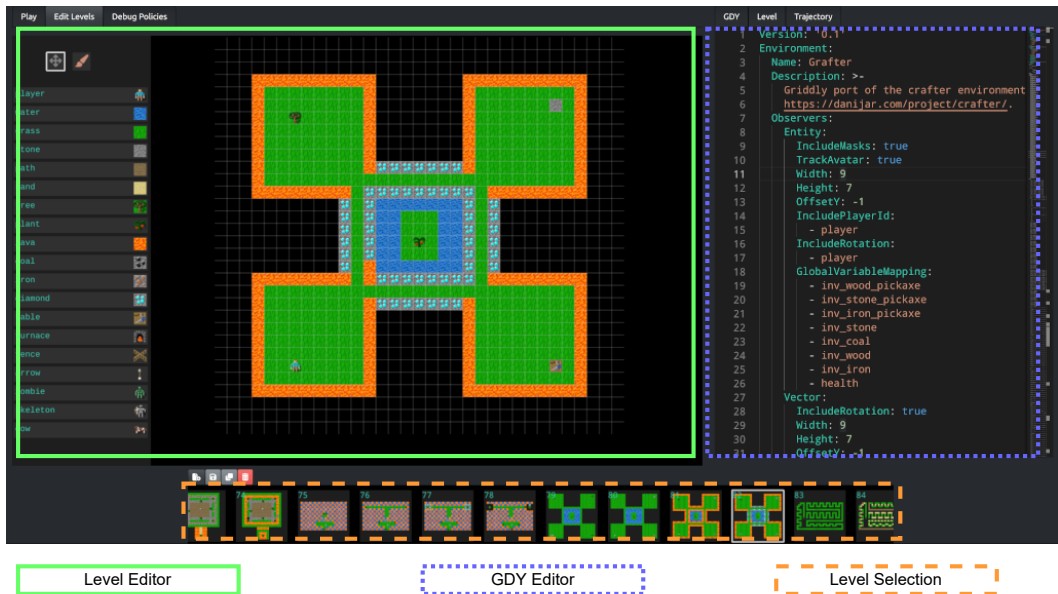

Figure 3: GriddlyJS has three main components: The level editor allows rapid design of custom levels with a code-free, visual interface; rendered levels are fully interactive via keyboard control. The GDY editor allows editing of the underlying GDY specification of the core environment mechanics. The level selection component lists previously designed levels. Users can select levels for further modification or deletion.

Griddly, *observers* are functions that determine how any given environment state is rendered into an observation. Several different types of observers are supported, including those shown in Figure 2. Griddly additionally supports lightweight observers based on default 2D block sprites, ASCII, and semantic entity states (as defined in the GDY). Observers are highly customizable, with options for partial observability, rotation, cropping, and inclusion of global or internal, entity-level variables. We provide a simple example Griddly environment implementation in Appendix A.

## 3  GriddlyJS

GriddlyJS provides a fully web-based integrated development environment (see Figure 3) composed of simple and intuitive user interfaces wrapping the core components of the Griddly engine. As such, the GriddlyJS IDE can be used inside any modern browser, without the need for installing complex dependencies. Running Griddly directly inside of the browser is made possible through transcompiling the core components of Griddly into WebAssembly (Wasm). The interface itself is written using the React library. Inside the GriddlyJS IDE, GDY files can be directly edited with any changes to the environment's mechanics immediately reflected. Moreover, specific levels of the

environment can be designed using a simple visual editor, allowing levels to be drawn directly inside the IDE. Previously designed levels can be saved locally into a gallery and instantly reloaded into the environment instance running inside the IDE and played via keyboard controls. Taken together, the features of GriddlyJS allow for rapid environment development, debugging, and experimentation. We now discuss the major highlights of GriddlyJS in turn. For a detailed walkthrough of these features, see Appendix B.

**Environment Specification** Designing new environment dynamics can be time-consuming. For example, adding and testing a new reward transition requires recompiling the environment and adding specific test cases. With GriddlyJS changes can be coded directly inside the GDY in the browser, where it will be immediately reflected in the environment. The designer can then interactively control the agent to test the new dynamic. Moreover, the environment's action space is automatically reanalyzed on all changes and environment actions are assigned sensible key combinations, e.g. WASD for movement actions. Similarly, newly defined entities inside the GDY are immediately reflected in the visual level editor, allowing for rapid experimentation.

**Level Design** Given any GDY file, GriddlyJS provides a visual level editor that allows an end user to design environment levels by drawing tiles on a grid. Objects from the GDY file can be selected and placed in the grid by pointing and clicking. The level size is automatically adjusted as objects are added to the it. The corresponding level description string, which is used by the Griddly environment in Python to reset to that specific level, is automatically generated based on the character-to-entity mapping defined in the GDY. New levels can then be saved to the same GDY file and loaded inside the Python environment.

**Publish to the Web** As GriddlyJS is built using the React library, the environment component itself can be encapsulated inside a React web component. Moreover, GriddlyJS supports loading and running of TF.js models directly inside the IDE environment instance. Taken together, this allows publishing of Griddly environments and associated agent policies in the form of TF.js models directly to the web, as an embedded React web component. By allowing researchers to directly share interactive demos of their trained agents and environments, GriddlyJS provides a simple means to publish reproducible results, as well as research artifacts that encourage the audience to further engage with the strengths and weaknesses of the methods studied.

**Recording Human Trajectories** Recording human trajectories for environments is as simple as pressing a record button in the GriddlyJS interface and controlling the agent via the keyboard. Recorded trajectories are saved as JSON, consisting of a random seed for deterministically resetting the environment to a specific level and the list of actions taken. They can easily be compiled to datasets, e.g. for offline RL or imitation learning. The recorded trajectories can also be replayed inside of GriddlyJS.

**Policy Visualization** Visualizing policy behavior is a crucial debugging technique in RL. Policy models can be loaded into GriddlyJS using TF.js and run in real-time on any level. In this way, the strengths, weaknesses, and unexpected behaviors of a trained policy can be quickly identified, providing intuition and clues about any bugs or aspects of the environment that may be challenging in a closed-loop development cycle. These insights can then be used to produce new levels that can bridge the generalization gap and thus improve the robustness of the agent.

## 4 Proof-of-Concept: Escape Room Puzzles

We now demonstrate the utility of GriddlyJS by rapidly creating a complex, procedurally-generated RL environment from scratch. After developing this new environment, we then use GriddlyJS to quickly hand-design a large, diverse collection of custom levels, as well as record a dataset of expert trajectories on these levels, which can be used for offline RL. We then load this new environment into Griddly for Python to train an RL agent on domain randomized levels—whose generation rules are defined within the associated GDY specification—and evaluate the agent's performance on the human-designed levels.

### 4.1 Rapid Environment Development

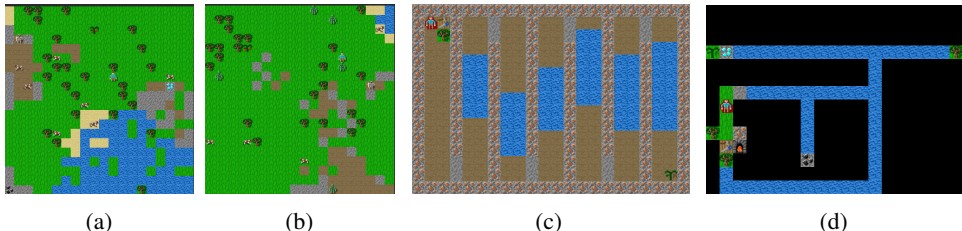

|     (a)     |     (b)     |     (c)     |     (d)     |

Figure 5: Example `EscapeRoom` that are procedurally-generated (a, b) and human-designed (c,d). The agent must collect resources to build tools and structures to reach the goal cherry tree, while surviving the environment.

We consider an environment, resembling a 2D version of MineCraft, in which the agent must learn a set of skills related to gathering resources and constructing entities using these resources in order to reach a goal. While prior works have presented environments with similar 2D, compositional reasoning challenges [3, 64, 66], we specifically model our `EscapeRoom` environment after the complex state and transition dynamics of Crafter [23], with the key difference being that `EscapeRoom` episodes terminate and provide a large reward upon reaching the goal object, a cherry tree, in each level.[3] The dynamics inherited from Crafter entail harvesting raw resources such as wood and coal in order to build tools like furnaces, bridges, and pickaxes required to harvest or otherwise clear the path of additional resource tiles like iron and diamond.

Mastering this environment presents a difficult exploration problem for the agent: Not only must the agent reach a potentially faraway goal, but it must also learn several subskills required to reliably survive and construct a path leading to this goal. Success in this environment thus requires exploration, learning modular subskills, as well as generalization across PCG levels. Mean-

```
- Name: do
  ...
  Behaviours:
    ...
    - Src:
        Object: player
        Preconditions:
          ...
        Commands:
          - add:
              - inv_wood
              - 1
          - if:
              Conditions:
                lt:
                  - ach_collect_wood
                  - 1
              OnTrue:
                - set:
                    - ach_collect_wood
                    - 1
                - reward: 1
      Dst:
        Object: tree
        Commands:
          - remove: true
          - spawn: grass
```

Figure 4: GDY for an environment transition for picking up wood.

while, implementing these rich dynamics presents a time-consuming challenge for the researcher. Such a complex environment typically entails knowledge of many disparate modules performing functions ranging from GPU-accelerated graphics rendering and vectorized processes for parallelized experience collection. Further, the researcher must implement complex logic for executing the finite-state automata underlying the environment transitions, as well as that handling the rendering of observations.

GriddlyJS abstracts away all of these details, allowing the researcher to focus exclusively on defining the underlying MDP through a succinct GDY specification. In particular, the researcher can simply define all entities (i.e. Griddly `Objects`) present in the game, each with an array of internal state variables, as well as the agent's possible actions. Then, all transition dynamics are simply established by declaring a series of local transition rules (i.e. Griddly actions) based on the state of each entity in each tile, as well as any destination tile, acted upon by the agent's action. For example, after declaring the action of `do` (i.e. interact with an object) along with the possible game entities and their states (e.g. the agent is the `player`, which can be `sleeping`) we can simply define the transition dynamic of receiving +1 wood resource upon performing `do` on a `tree` using the simple sub-block declaration shown in Figure 4. More complex dynamics can be implemented by calling built-in algorithms like A* search or nesting Griddly `Action` definitions. Further, arbitrary PCG logic can be easily implemented by writing Python subroutines that output level strings corresponding to the ordering of the level tiles.

### 4.2 Human-in-the-Loop Level Design

Given the rich design space of the `EscapeRoom` environment, randomized PCG rules defined by the Griddly level generator are unlikely to create challenging levels that push the boundaries of the agent's current capabilities. Rather in practice, designing such challenging levels for such puzzle games

---

[3]A full description of how `EscapeRooms` deviates from Crafter is provided in Appendix C.

rely heavily on human creativity, intuition, and expertise, which can quickly hone in on the subsets of levels posing unique difficulties for a player or AI. Indeed, based on recent works investigating the out-of-distribution (OOD) robustness of RL agents trained on domain-randomized (DR) levels [17, 29], we do not expect agents trained purely on randomized PCG levels to perform well on highly out-of-distribution, human-designed levels, without the usage of such adaptive curricula. However, it can be costly to collect a large set of diverse and challenging human-designed levels necessary to encompass the relevant challenges, and thus, most prior works test on limited sets of human-designed OOD levels.

GriddlyJS allows us to quickly assess OOD generalization on a large number of human-designed levels. With its visual level editor and interactive, browser-based control of agents, we can rapidly design and iterate on new and challenging levels. In particular, we created 100 diverse environments in roughly eight hours, many featuring environment and solution structures that are highly unlikely to be generated at random. We then use PPO to train a policy on domain-randomized `EscapeRoom` levels. We checkpoint the policy at regular intervals in terms of number of PPO updates and evaluate the performance of each checkpoint on all 100 human-designed levels, as well

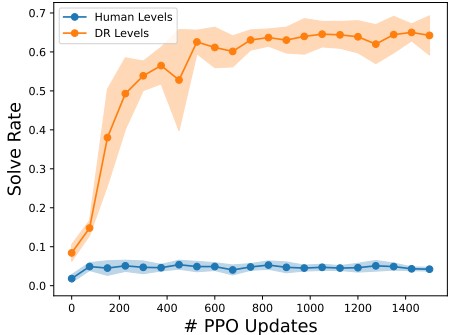

Figure 6: Mean and std of solve rate on DR levels (orange) and human-designed levels (blue).

as 100 DR levels. We see in Figure 6 that, throughout training, the resulting policy solves DR levels at a significantly higher rate than human-designed levels, highlighting the distinct quality of human-designed levels. We also note here that the human-designed levels are exponentially unlikely to fall inside the distribution of the generated levels. This highlights a significant limitation of PCG, which does not exist with levels generated by a human. Generating complex puzzle levels using PCG methods is an ongoing area of research. Tuning generators to create unique levels often results in levels that are invalid or unsolvable, and vice versa [57, 63, 54, 18, 17]. The full details on our choice of model architecture and hyperparameters is provided in Appendix C.

Furthermore, as GriddlyJS loads TF.js models for policy evaluation and visualization directly inside the IDE, human-in-the-loop level design can be performed in a closed-loop, adversarial manner: The policy, first trained on DR levels, is successively evaluated on additional sets of human-designed levels, the most challenging of which are added to the agent's training set, thereby robustifying the agent's weaknesses. Given the success of these methods and that of recent adversarial adaptive curricula methods for RL [30] in producing robust models, we expect human-in-the-loop adversarial training to lead to similarly significant gains in policy robustness. Importantly, by developing an RL environment in GriddlyJS, this mode of training is immediately made available to the researcher. GriddlyJS enables TF.js policies and environments to be directly published on the web, thus allowing such adversarial methods to be tested at high scale, potentially leading to highly robust policies and collecting unique datasets of adversarial levels useful to future research in generalization and the emerging field of unsupervised environment design [17].

## 4.3 Recording and Controlling Trajectories

GriddlyJS makes it easy to record trajectories for any level directly inside the IDE, enabling a wide range of downstream use cases. For example, such recorded trajectories can be associated with human-designed levels during human-in-the-loop adversarial training to ensure solvability. Further, such recorded trajectories naturally serve as datasets for offline RL and imitation learning—especially useful for more complex multi-task or goal-conditioned environments where it may be important to ensure the dataset has sufficient coverage of the various tasks or goals [41].

Moreover, the interactive control feature in the presence of a loaded TF.js policy enables the study of human-AI interaction, in which the agent may hand over control to a human when the policy is uncertain. Further, as levels can be edited directly inside the IDE, GriddlyJS allows researchers to perform controlled evaluation of policy adaptation to environment changes that occur mid-episode.

# 5   Related Works

Several systems for developing custom procedurally-generated environments have been introduced, taking the form of code libraries featuring a domain-specific language (DSL) and encompassing both grid worlds [50, 42, 31, 7], including the Griddly library for Python [6], as well as more complex, physics-based environments [51, 32]. More recently, MiniHack [49] builds on top of the NetHack Learning Environment [NLE; 35] to provide a library for developing grid worlds in the NetHack runtime. These systems focus exclusively on programmatic environment creation and provide no additional functionality for design, debugging, and agent evaluation.

Other related systems include the ML-Agents module, which converts Unity games into RL environments [32]. The resulting trained agents can then be loaded back into Unity for evaluation. However, ML-Agents provides no specific tooling for constructing environments. Recent works have also provided interactive evaluation interfaces, similar to that provided by GriddlyJS. TeachMyAgent [48, 22] provides a user interface for visualizing the performance of pre-trained RL agents in a limited set of 2D continuous-control tasks extending those in OpenAI Gym [10]. Other recent works have provided interactive, web-based demonstrations of pre-trained agents in PCG environments [40, 46]. Unlike GriddlyJS, they focus on interactive visualization, rather than providing a streamlined, closed-loop workflow connecting environment development to agent training, evaluation, and publication via the web. Such experiments in publishing continue a rich lineage of works exploring interactive articles [26], a mode of publication whose popularity GriddlyJS seeks to catalyze.

PCG has enabled many findings on RL robustness and generalization, including the important, preliminary result that RL methods easily overfit to singleton environments [20, 39, 65]. Since then, many methods for improving robustness and generalization of RL agents have been studied, notably various forms of policy regularization, data augmentation strategies [45, 62], model architecture [36, 9], feature distillation [28, 13, 44], and adaptive curricula [43, 30, 29, 17, 40]. Still, the number of available PCG environments is ultimately limited to a mostly arbitrary selection determined by the interests of the select few research groups who have invested the resources to produce these environments. As such, this fast-growing area of RL research risks overfitting to a small subset of environments. By streamlining PCG environment creation, agent evaluation, and interactive result sharing, GriddlyJS seeks to empower a the wider research community.

# 6   Conclusions and Future Work

We introduced GriddlyJS, a fully web-based integrated development environment that streamlines the development of grid-world reinforcement learning (RL) environments. By abstracting away the implementation layers responsible for shared business logic across environments and providing a visual interface allowing researchers to quickly prototype environments and evaluate trained policies in the form of TensorFlow.js models, we believe GriddlyJS can greatly improve research productivity. Moreover, in future work, we plan to use web-based tracking of anonymized user behaviors (e.g. those correlated with notions of productivity, like number of environments created and mean time for environment creation) and standard A/B testing methods to better understand how GriddlyJS is used, track bugs, and surface usability pain-points. Such a data-driven approach allows us to iteratively improving GriddlyJS for users based on direct quantifications of user productivity. GriddlyJS enables human-in-the-loop environment development, which we believe will become a major paradigm in RL research and development, allowing for the measured design of higher quality environments and therefore training data. Moreover, such approaches (and therefore GriddlyJS) can enable new training regimes for RL, such as human-in-the-loop adversarial curriculum learning.

Of course, our system is not without limitations: GriddlyJS does not currently persist user-generated data on a dedicated server, though we plan to support this functionality in the future. Additionally, although environments are rendered in the browser, pixel-based observation states are not currently supported. Moreover, training must still occur outside of GriddlyJS, a bottleneck that is mitigated by the fact that GDY files can be so easily loaded into Griddly for Python, and most model formats are easily converted to the TF.js format as highlighted in Appendix B.4.

As a proof-of-concept, we used GriddlyJS to rapidly develop the `EscapeRoom` environment based on the complex skill-based dynamics of Crafter, along with 100 custom hand-designed levels. We then demonstrated that an agent trained on domain-randomized levels performs poorly on human-designed

ones. This result shows that PCG has difficulty generating useful structures for learning behaviors that generalize to OOD human-designed levels, thereby highlighting the value of GriddlyJS's simple interface for quickly designing custom levels. Additionally, we believe GriddlyJS's web-first approach will enable more RL researchers to share their results in the form of interactive agent-environment demos embedded in a webpage, thereby centering their reporting on rich and engaging research artifacts that directly reproduce their findings. Taken together, we believe the features of GriddlyJS have the potential to greatly improve the productivity and reproducibility of RL research.

## 7 Acknowledgements

Firstly, we would like to thank Meta for funding this research and the AI Creative Design team and Quiddale O'Sullivan for the "Griddly Bear" logo. Secondly, we would like to thank the following people for testing and feedback on GriddlyJS: Remo Sasso, Sebastian Berns, Elena Petrovskaya and Akbir Khan. Finally, we are grateful to anonymous reviewers for their recommendations that helped improve the paper.

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
