# A Griddly GDY Overview

The Game Description YAML (GDY) is the domain-specific language of the Griddly framework that provides a means to easily design rich and diverse grid-based environments. Based on the Yet Another Markup Language (YAML), this human-readable language features simple declarative rules. GDY files are composed of three main sections, namely `Environment`, `Actions`, and `Objects`.

To provide an intuition on how environments can be implemented using Griddly, we provide a brief tutorial of the GDY below which recreates the popular game of Sokoban. Once the GDY is completed, Griddly's environment wrappers can easily map it to a Gym environment.

## A.1 Objects

In this section we describe the objects of the game and the ways they can be rendered in Griddly.

The game of Sokoban is composed of four objects: **avatar**, **box**, **hole**, and **wall**. **avatar** is the main decision-making object which can move around and push boxes into holes. Walls are immovable objects. The goal of the **avatar** is to push all boxes into the holes.[4]

Each object needs to have a unique name, which can serve as references to that object in other parts of the GDY code. Let us start the `Objects` section and define the **avatar** object as follows:

```
Objects:
 - Name: avatar
   Z: 2
   MapCharacter: A
   Observers:
     Sprite2D:
       Image: images/gvgai/oryx/knight1.png
```

`MapCharacter` is used to define the ASCII character of an object and can be used to mark the initial positions of objects in concrete levels defined later in the `Environment` section.

The property $Z$ can serve as the third dimension of the cells in the grid. It allows to define objects that can occupy the same location of the grid, as long as they have different $Z$ values. It also defines the order of objects when rendered in Griddly, i.e., higher $Z$ values indicate that the objects will be rendered on top.

The `Observers` property determines how each observer type will render this particular object. Here, the **avatar** object only includes a Sprite2D observer, but Griddly supports additional forms of observations, including those shown in Figure 2. Sprite2D observers expect an image for rendering, thus we select a knight icon from Griddly's selection of icons.[5]

Having finished the description of the **avatar** object, we proceed to the **wall** object. Here we provide 16 different images for walls to correspond to different positions of walls, such as horizontal or vertical locations, corner pieces, T-pieces, etc.

```
- Name: wall
  MapCharacter: w
  Observers:
    Sprite2D:
      TilingMode: WALL_16
      Image:
        - images/gvgai/oryx/wall3_0.png
        - images/gvgai/oryx/wall3_1.png
        - images/gvgai/oryx/wall3_2.png
        - images/gvgai/oryx/wall3_3.png
        - images/gvgai/oryx/wall3_4.png
        - images/gvgai/oryx/wall3_5.png
        - images/gvgai/oryx/wall3_6.png
```

---

[4]There are many variations of the game of Sokoban. In our particular implementation the agent can push boxes into any hole on the environment. There also exist versions where only a single box can be pushed into each hole.

[5]Griddly allows users to easily upload new custom icons for their own environments.

```
     - images/gvgai/oryx/wall3_7.png
     - images/gvgai/oryx/wall3_8.png
     - images/gvgai/oryx/wall3_9.png
     - images/gvgai/oryx/wall3_10.png
     - images/gvgai/oryx/wall3_11.png
     - images/gvgai/oryx/wall3_12.png
     - images/gvgai/oryx/wall3_13.png
     - images/gvgai/oryx/wall3_14.png
     - images/gvgai/oryx/wall3_15.png
```

We do not provide this object with a Z value given that nothing should interact with this object. WALL_16 tiling mode is used to make sure all 16 wall icons are rendered correctly for each location.

```
- Name: box
  Z: 2
  MapCharacter: b
  Observers:
    Sprite2D:
      Image: images/gvgai/newset/block1.png

- Name: hole
  Z: 1
  MapCharacter: h
  Observers:
    Sprite2D:
      Image: images/gvgai/oryx/cspell4.png
```

The **box** and **hole** objects can be defined similar to the **avatar** objects, except that the **hole** objects have a different Z value allowing the avatar to move on top of them.

## A.2   Actions

Actions define the mechanics of the game and interactions between objects in Griddly. Each individual action includes two entities: **source** and **destination**. **source** is the object which performs a particular action, whilst the **destination** is the object that is affected by this action. Firstly, we define the movement action of the **avatar** as follows.

```
Actions:
 # Define the move action
 - Name: move
   Behaviours:
   # The agent can move around freely in empty space and over holes
     - Src:
         Object: avatar
         Commands:
           - mov: _dest
       Dst:
         Object: _empty
```

Given that the Src key includes **avatar** object as its value, it can be inferred that this is an action performed by the **avatar**. The Dst key with the object value _empty indicates that the behaviour only applies when the action is performed on a space with no objects on it.

The Commands property in the Src field includes a list of instructions that will be executed to the Src object once this action is performed. The mov: _dest commands moves the object to the destination of the action.

Next, we define the box pushing actions. Firstly, we define the ability of **box** objects to move to empty locations. Then, we allow the **avatar** object to interact with the **box** object.

```
# Boxes can move into empty space
- Src:
    Object: box
    Commands:
        - mov: _dest
```

```
  Dst:
    Object: _empty

# The agent can push boxes
- Src:
    Object: avatar
    Commands:
        - mov: _dest
  Dst:
    Object: box
    Commands:
        - cascade: _dest
```

Here we make sure that the **box** object is moved in the same direction as the **avatar** object, the source of the action. We achieve this by using the `cascade: _dest` which re-apply the same action on the destination object, namely the **box**.

Finally, we define the mechanics of the **box** being pushed onto a **hole**. We achieve this by defining our last action with **box** as its source and **hole** as its destination:

```
# If a box is moved into a hole remove it
 - Src:
    Object: box
    Commands:
      - remove: true
      - reward: 1
  Dst:
    Object: hole
```

Here, the `remove: true` command removes the source **box** from the grid once pushed into a hole. Furthermore, the `reward: 1` commands Griddly to provide the agent with the reward of 1 once this event is triggered.

### A.3 Environment

The `Environment` section defines the environment description, such as its name, as well as the observation and action spaces of the MDP. For all environments, Griddly uses the canonical action space layout from [5], with support for invalid action masking. In Griddly, *observers* are functions that determine how any given environment state is transformed into an observation. Several different types of observers are supported, including those shown in Figure 2. Griddly additionally supports lightweight observers based on default 2D block sprites, ASCII, and semantic entity states (as defined in the GDY). Observers are highly customizable, with options for partial observability, rotation, cropping, and inclusion of global or internal, entity-level variables.

Below we provide the `Environment` section for our Sokoban example. Firstly, we indicate that the **avatar** object will serve as the decision-making agent in our environment:

```
Player:
  AvatarObject: avatar
```

We then describe the termination condition which determines when the episode is complete and whether the agent wins or loses. For Sokoban, an episodes is considered won if all the boxes are pushed into the holes, i.e., the number of boxes in the environment is equal to 0:

```
Termination:
    Win:
     - eq: [box:count, 0]
```

Our next step is to defines levels for our Sokoban game. The layout of each level can be defined using a sequence of string that uses the `MapCharacter` characters of each object defined above. The dot character . indicates an unoccupied grid cell.

```
Levels:
    - |
```

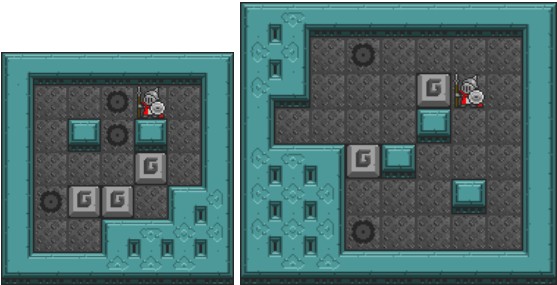

Figure 7: Custom Sokoban levels defined in the GDY example.

```
    wwwwwww
    w..hA.w
    w.whw.w
    w...b.w
    whbb.ww
    w..wwww
    wwwwwww
  - |
    wwwwwwwww
    ww.h....w
    ww...bA.w
    w....w..w
    wwwbw...w
    www...w.w
    wwwh....w
    wwwwwwwww
```

The two defined levels produce the environment renderings illustrated in Figure 7.

Lastly, we specify the size of the tiles in pixels and the background image using the `MapCharacter` and `BackgroundTile` fields. We also provide the environment with a unique name.

```
Environment:
    Name: sokoban
    TileSize: 24
    BackgroundTile: images/gvgai/newset/floor2.png
```

## A.4  Putting It All Together

Figures 8 and 9 provide the full implementation of the Sokoban example in Griddly.

```
Environment:
    Name: sokoban
    TileSize: 24
    BackgroundTile: images/gvgai/newset/floor2.png
    Player:
      AvatarObject: avatar
    Termination:
      Win:
        - eq: [box:count, 0]
    Levels:
      - |
        wwwwwww
        w..hA.w
        w.whw.w
        w...b.w
        whbb.ww
        w..wwww
        wwwwwww
      - |
        wwwwwwwww
        ww.h....w
        ww...bA.w
        w....w..w
        wwwbw...w
        www...w.w
        wwwh....w
        wwwwwwwww

Actions:
- Name: move
  Behaviours:
    - Src:
        Object: avatar
        Commands:
          - mov: _dest
      Dst:
        Object: [_empty, hole]

    - Src:
        Object: box
        Commands:
            - mov: _dest
      Dst:
        Object: _empty

    - Src:
        Object: avatar
        Commands:
          - mov: _dest
      Dst:
        Object: box
        Commands:
          - cascade: _dest

    - Src:
        Object: box
        Commands:
          - remove: true
          - reward: 1
      Dst:
        Object: hole
```

Figure 8: Full implementation of Sokoban in Griddly (part 1).

```
Objects:
 - Name: box
   Z: 2
   MapCharacter: b
   Observers:
     Sprite2D:
       Image: images/gvgai/newset/block1.png

 - Name: wall
   MapCharacter: w
   Observers:
   Sprite2D:
     TilingMode: WALL_16
     Image:
       - images/gvgai/oryx/wall3_0.png
       - images/gvgai/oryx/wall3_1.png
       - images/gvgai/oryx/wall3_2.png
       - images/gvgai/oryx/wall3_3.png
       - images/gvgai/oryx/wall3_4.png
       - images/gvgai/oryx/wall3_5.png
       - images/gvgai/oryx/wall3_6.png
       - images/gvgai/oryx/wall3_7.png
       - images/gvgai/oryx/wall3_8.png
       - images/gvgai/oryx/wall3_9.png
       - images/gvgai/oryx/wall3_10.png
       - images/gvgai/oryx/wall3_11.png
       - images/gvgai/oryx/wall3_12.png
       - images/gvgai/oryx/wall3_13.png
       - images/gvgai/oryx/wall3_14.png
       - images/gvgai/oryx/wall3_15.png

 - Name: hole
   Z: 1
   MapCharacter: h
   Observers:
     Sprite2D:
       Image: images/gvgai/oryx/cspell4.png

 - Name: avatar
   Z: 2
   MapCharacter: A
   Observers:
     Sprite2D:
       Image: images/gvgai/oryx/knight1.png
```

Figure 9: Full implementation of Sokoban in Griddly (part 2).

## B  GriddlyJS UI Walkthrough

In this section we show various screenshots of the GriddlyJS user interface and highlight various useful features.

### B.1  Building And Testing Environment Mechanics

GriddlyJS provides many tools for building and debugging environments. Figure 10 shows several of these. Firstly, as soon as GDY file is loaded in the editor and a level selected, it will be playable in the editor window. Actions in the environment are automatically mapped to the keyboard, and an explanation of this mapping can be toggled by pressing **P**. Additionally all global and player-wise variables can be toggled by pressing **I**. These variables are updated live while the game is being played.

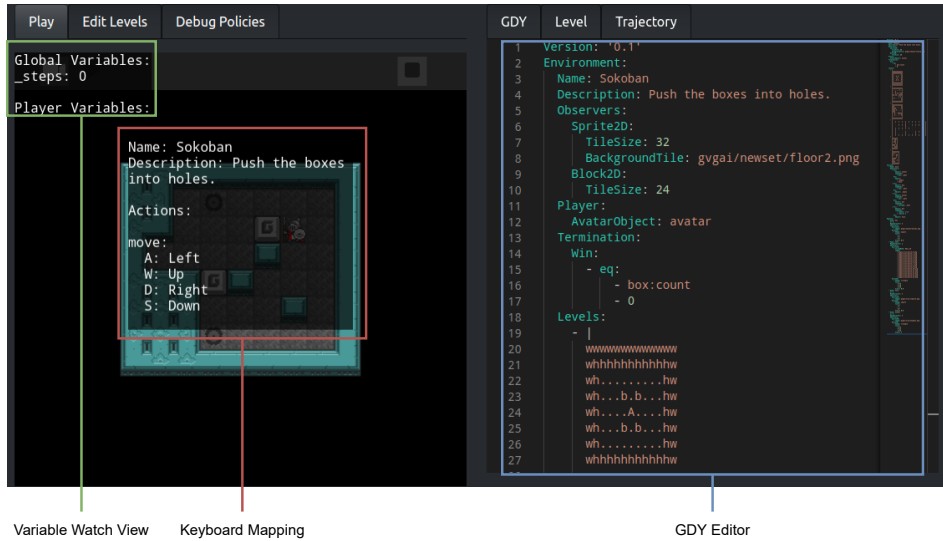

Figure 10: Environment Designing and Debugging interfaces..

## B.2   Level Design

The Level Editor view shown in Figure 11 allows the user to choose objects to place on the game grid in order to create levels. The user can selects an object from the menu and then can *paint* it into the editing grid. The editing grid automatically grows if objects are placed near the boundaries, so levels of any shape or size can be created. Additionally as objects are painted into the editor grid, a level string is automatically generated and displayed. This level string can also be manually edited and its changes reflected in the editor window.

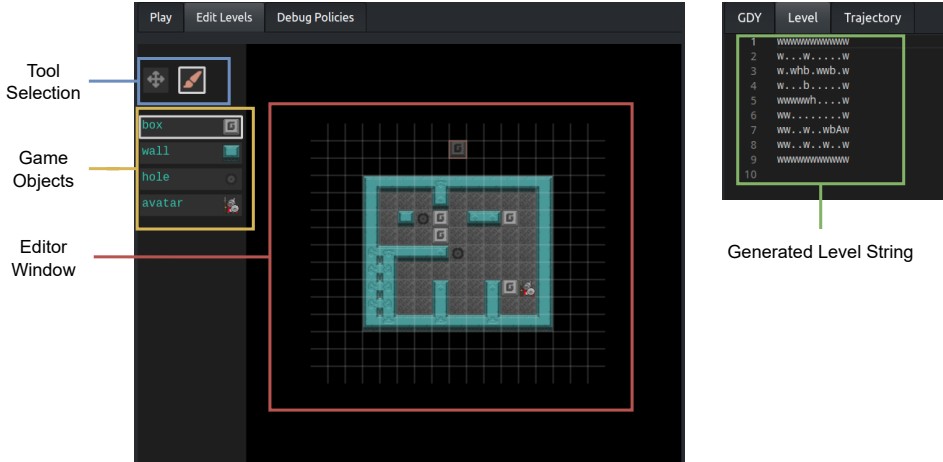

Figure 11: Level Editor view and level string view..

Managing the set of levels in the environments GDY file is handled by the Level Selection interface shown in Figure 12. Users can create, update and delete levels in the GDY file for quickly generating large datasets of levels.

## B.3   Recording Trajectories

Recording and playback options in the GriddlyJS interface. On clicking the **Record** icon, the user's actions are recorded as they play the game in the environment view. When the environment terminates, it is reset to its original state and a **play** button is shown next to the record button. If pressed, the

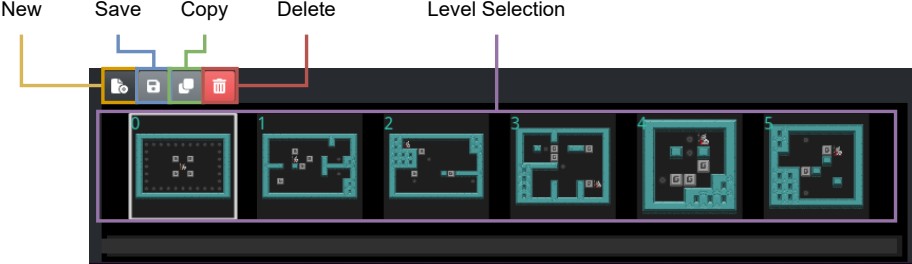

Figure 12: The level selector view showing thumbnails of the the levels in the GDY..

play button re-plays the recorded trajectory. These options are shown in Figure 13. Additionally, the actions and seed are displayed as YAML in the trajectory view in the editor. This trajectory can then be copied and stored for later use, for example in behavioural cloning algorithms. Trajectories can also be copied into the text-area from external sources and played within the editor.

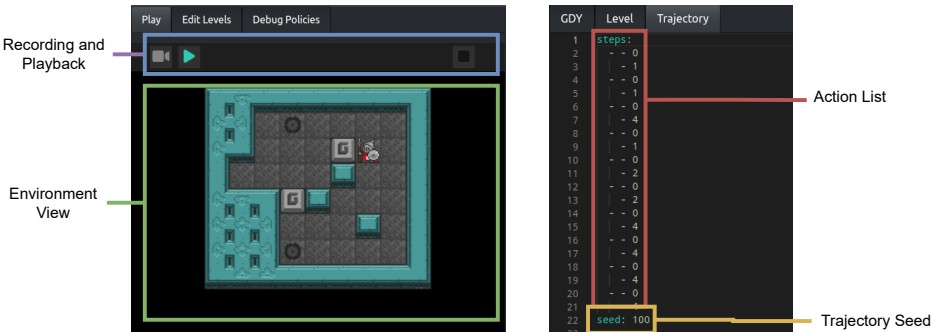

Figure 13: Recording and playback menus for generating and viewing trajectories.

## B.4 Evaluating Models

Trained policies can be loaded into GriddlyJS and replayed using the Debug Policies view, shown in Figure 14. If a model is loaded, a **play** button will be visible which will sample actions from the policy to view its performance. once the episode is finished, the level is reset.

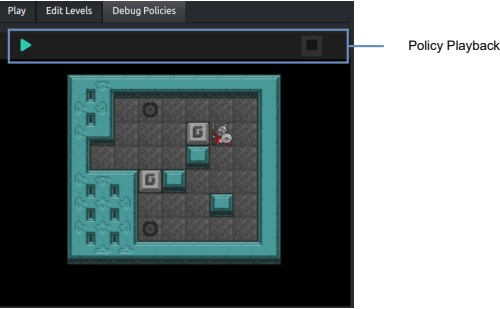

Figure 14: Policy debugger view, similar to the "play" view but only visible if a policy is loaded using TensorflowJS.

### B.4.1 Deep Learning Framework Support

GriddlyJS supports any deep learning model that can be converted into the ONNX format [1]. This includes many popular frameworks such as **PyTorch**, **Tensorflow**, **JAX**, **Caffe** and **Chainer**. Once converted to the ONNX format, these models can be converted to TensorflowJS and used in the debugging view.

As our experiments in section 4 are trained using PyTorch, we include example scripts to convert these models to ONNX and then to TensorflowJS. These scripts and documentation on how to load and use the converted models can be found at: `https://github.com/GriddlyAI/escape-rooms#using-checkpoints-in-griddlyjs`

## C   Experimental Details and Hyperparameters

Table 1 summarises the hyperparameters we chose to sweep. Other hyperparameters while sweeping were those shown in Table 2.

Table 2 summarises our final hyperparameter choices for our PPO agent.The final choice was made by taking highest average (calculated across the seeds) level completion rate.

Table 1: Hyperparameter sweep values

| Parameter | Values |
| --- | --- |
| $\lambda_{\text{GAE}}$ | 0.65, 0.8, 0.95 |
| Adam learning rate | 5e-2, 1e-2, 5e-3, 1e-3, 5e-4, 1e-4 |
| Student entropy coefficient | 0.2, 0.1, 5e-2, 1e-2, 5e-3, 1e-3 |
| Seeds | 0 1 2 3 4 5 6 7 8 9 |

Table 2: Hyperparameters used for training the PPO model.

| Parameter | Values |
| --- | --- |
| $\gamma$ | 0.99 |
| $\lambda_{\text{GAE}}$ | 0.95 |
| PPO rollout length | 128 |
| PPO epochs | 4 |
| PPO minibatches per epoch | 4 |
| PPO clip range | 0.2 |
| PPO number of workers | 256 |
| Adam learning rate | 1e-3 |
| Adam $\epsilon$ | 1e-5 |
| PPO max gradient norm | 0.5 |
| PPO value clipping | yes |
| Return normalization | no |
| Value loss coefficient | 0.5 |
| Student entropy coefficient | 0.05 |

### C.1   Architecture

We use the PPO implementation from CleanRL [27] with the *ImpalaCNN* [19] architecture as this is commonly used with grid-world environments.

### C.2   Training And Evaluation

All training and evaluation episodes are limited to 500 steps. The agent receives no penalty for reaching this limit. We trained our models with 10 different seeds for 50 million environment steps. All training is performed using our modified Crafter level generator as described in the next section. All results are averaged across these 10 seeds. All code for our experiments and descriptions on how to use the training and evaluation scripts can be found in the escape-rooms repository: `https://github.com/GriddlyAI/escape-rooms`

### C.3 Modified Crafter Environment

Griddly's GDY format contains and restricted set of commands that allow complex mechanics to be realised. However when translating from many environments into GDY format, there are some caveats that may mean behaviours are slightly modified from the original versions.

#### C.3.1 Grafter

Grafter Github repository: `https://github.com/GriddlyAI/grafter`

Before generating the Escape Room environments, Crafter was first translated directly to GDY to create as close a replication of the original environment as possible. This replication (Nicknamed Grafter) had several features that could not be directly translated. These translation artifacts between the environment implementations are explained below:

**Chunk Balancing** The spawning and despawning of Non-player characters (NPC) i.e zombies, cows, and skeletons in order to balance their numbers across the environment is not possible using the current features of Griddly, so objects are only spawned at the start of the episode. Defeating NPCs removes them permanently from the environment.

**Day and Night Cycles** Changing the brightness of pixels in order to simulate a day and night cycle is relevant only when pixel observations are being used. Griddly supports several other observation spaces where day and night cannot be easily modelled, such as `Vector` and `Entity`. Day and night is still implemented as part of the `Sprite2D` observations, but the associated behavioural changes for NPCs are not present.

**Chasing Behaviour** Zombies and skeletons use Griddly's built-in A* pathfinding implementation, whereas zombies and skeletons in Crafter use a simple rule-based method.

**Observation Spaces** All observation spaces are configured to contain the same information as the original crafter environment and have equivalent observability dimensions. **Sprite2D** observers configured to be the same as the original Crafter environment, the inventory display and day/night cycle are produced by a custom shader. In this implementation, the voronai pixel noise used in the original environment is ommited. **Vector** observers contain a 7x9x51 (WxHxC) observation space, where the channels $C$ represent object types, orientations, playerIds and the set of global variables which represent the inventory, which are repeated across the height and width dimensions. **Entity** observer contain a list of features for each object type in the 7x9 space around the agent and additionally include a *global* entity which contains the inventory variables.

**Multi-Agent Support** is naturally introduced as part of the features that come with Griddly, agents gain an additional achievement if they defeat other agents. The number of agents that are spawned in the environment is configurable in the GDY.

#### C.3.2 Domain Randomization

The domain randomization algorithm we use is a modified version of the open-simplex based level generator from the original Crafter environment. While the structure of the levels is generally the same as those in Crafter, we also make sure that we add a single "cherry" tree goal to each level. In most cases the cherry tree can be reached by traversing land, but occasionally there may be levels where more complex strategies are required such as chopping trees or building bridges to get to the island where a tree exists. We show examples of the DR levels in Figure 15

#### C.3.3 Escape Rooms

Escape Rooms Github repository `https://github.com/GriddlyAI/escape-rooms`

To make an escape room, firstly we needed a method of **escape**. To do this we repurposed the mechanic of eating a **plant** object. We added a termination condition so that if the *eat plant* goal is achieved, the episode ends and the agent receives a reward of 10.

There are also several features in Grafter that were not required in the Escape Room environment:

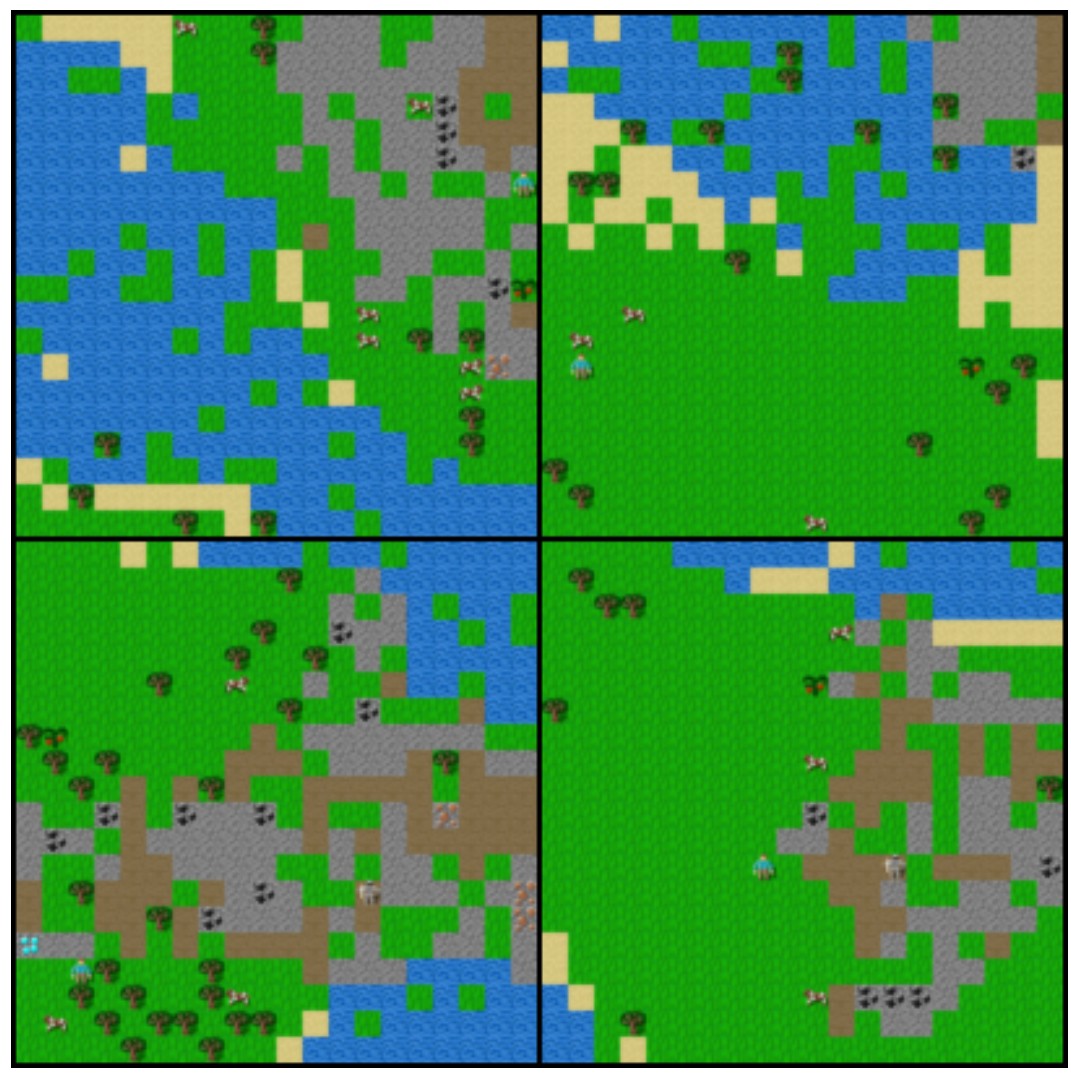

Figure 15: Example escape rooms generated by the Domain Randomization generator.

Table 3: Flattened Action Space

| Action | Values |
| --- | --- |
| No-Op | 0 |
| Move Left | 1 |
| Move Right | 2 |
| Move Down | 3 |
| Move Up | 4 |
| Interact With Object | 5 |
| Place Stone | 6 |
| Place Table | 7 |
| Place Furnace | 8 |
| Make Wood Pickaxe | 9 |
| Make Stone Pickaxe | 10 |
| Make Iron Pickaxe | 11 |

**Plants**  As reaching and eating a plant (cherry tree) is now being used as the goal state, the mechanics for collecting, planting and ripening of the trees was removed. Plants are spawned in the *ripe* state and remain that way until the agent collects them and subsequently ends the episode.

**Agent Survival** The mechanics for surviving in the environment, such as requiring food, water, energy and maintaining health levels are unnecessary complexities for the escape rooms and limit the possible challenges that can be built. The same reasoning is applied to zombies and skeletons which are not required. Removing these mechanics also removes the need for certain actions such as sleeping. Additionally, the day/night mechanics were removed entirely.

**Swords** Similar to the reasoning behind survival mechanics, we decided that combat with zombies/skeletons was an unnecessary complication, therefore the mechanics for building weapons were not required. This also simplified the action space as is shown in table 3.

**Reward Shaping** In Crafter the agent is rewarded depending on their current health level, as we are not using any health or survival mechanics this reward scheme is ommitted. All achievements still give a single reward of 1 for the first time they are encountered. The exception being the *eat plant* achievement which gives a reward of 10 for completing the escape room.

## D   Solution Trajectories

To demonstrate the ease of creating custom levels and recording trajectories with GriddlyJS IDE, we create a set of 100 hand-designed levels of the Crafter-based `EscapeRoom` environment. All 100 levels are visualized in Figure 22. The levels are feature distinct challenges for the agent. For each level, we include a solution trajectory generated by a human player using the recording feature inside the IDE. Griddly stores trajectories as simply a list of actions taken, along with a string representation of the level or a specific seed that allows the level generator to deterministically reset to the recorded level. We visualize key frames (left to right, top to bottom) from expert trajectories for a diverse subset of the 100 hand-designed `EscapeRoom` levels in Figures 16 through 21. In each frame, the agent is highlighted with a magenta bounding-box for clarity.

## E   Assets

In this section we outline the assets and open source software used in GriddlyJS, and any associated licenses.

**Oryx Design Lab** As Griddly is built as a successor to GVGAI for use in Reinforcement Learning, many of the assets used in GVGAI are used in Griddly. As the original assets in GVGAI are from the Oryx Design lab[6], we re-purchased the equivalent asset packs for use in Griddly as per the license agreement `https://www.oryxdesignlab.com/license`.

**GVGAI** Many of the Griddly environments are inspired by those in GVGAI. For example the levels and mechanics in games like Sokoban are clones of those in GVGAI. GVGAI is distributed under the following GNU GPL license: `https://github.com/GAIGResearch/GVGAI/blob/master/LICENSE.txt`

**Griddly** Griddly uses additional asset packs from the Oryx Design Lab as well as those used in GVGAI. A full list of asset packs that Griddly uses:

- **Iso Dungeon** `https://www.oryxdesignlab.com/products/iso-dungeon`

- **Tiny Galaxy** `https://www.oryxdesignlab.com/products/tiny-galaxy-tileset`

- **16-bit Fantasy** `https://www.oryxdesignlab.com/products/16-bit-fantasy-tileset`

As GriddlyJS is an extension on top of Griddly, we distribute GriddlyJS under the same MIT license: `https://github.com/Bam4d/Griddly/blob/develop/LICENSE`

**Crafter** As we clone the crafter environment, we also copy the assets used. Crafter and its assets are released under the MIT license: `https://github.com/danijar/crafter/blob/main/LICENSE`

---

[6]http://www.oryxdesignlab.com

# F   Broader Impact

GriddlyJS aims to drastically improve the productivity of RL research, by streamlining the pipeline from environment development to agent training and evaluation as a closed-loop workflow, and by enabling researchers to easily publish their findings as interactive agent-environment demos that invite active inquiry from the community. By furthering progress in RL, likely a major component of the most powerful AI systems of the future, GriddlyJS aligns with AI progress. Thus, our work aligns with the downsides of more rapid AI progress as well, namely the potentially faster proliferation of autonomous systems that may be put to malicious uses, such as the spread of misinformation and the deliberate or unintentional magnification of social and economic inequalities.

Furthermore, by simplifying the development of RL environments, the systematic biases in GriddlyJS's design may be amplified at scale, resulting in future RL research to overfit to these biases—the most obvious being GriddlyJS's deliberate, exclusive focus on grid world environments. Therefore, it is important to develop deeper understanding of the limitations of the grid world environments produced by GriddlyJS, for example, by conducting experiments comparing the computational efficiency and generalization properties of RL agents trained inside grid worlds compared to continuous control environments, under a diverse set of MDP formulations.



Figure 16: **Level 1.** In this simple level, the agent must solely navigate the maze to reach the goal cherry tree.

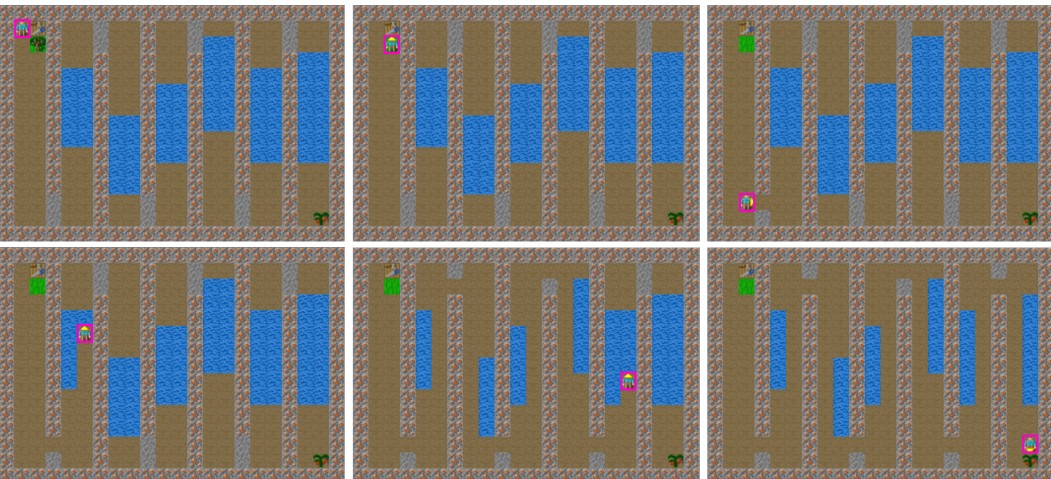

Figure 17: **Level 30.** The agent begins in the top-left corner, where it must first collect wood and build a wooden pickaxe using the work table. With this tool, the agent can pick up stones to clear the path, as well as successively place and pick up stones over the water to create a walkable path, making it possible to reach the goal.

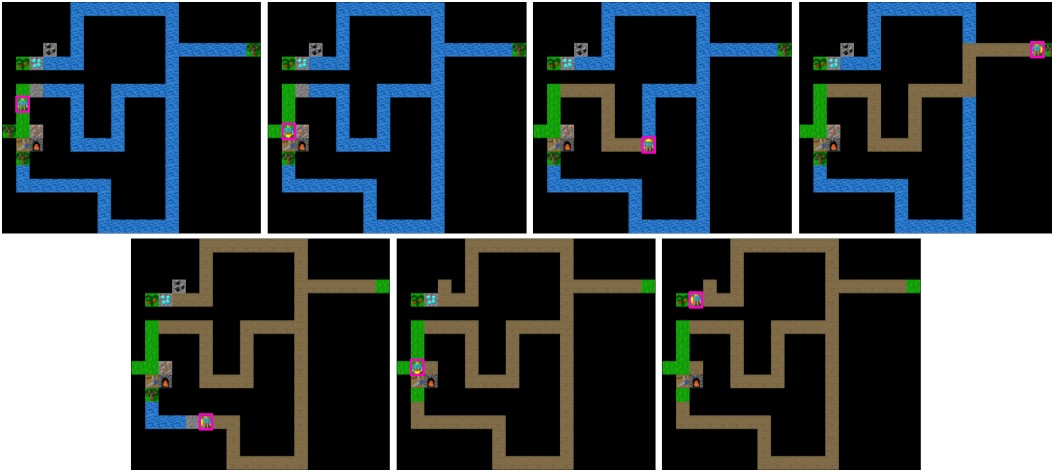

Figure 18: **Level 37.** The agent starts on the left and must first collect the wood and go to the work table to build a wooden pickaxe, with which it can collect the stone above. The agent must then successively place and remove this stone over the water to make the water walkable paths, making sure to collect the remaining two pieces of wood. Returning to the work table, the agent can then build a stone pickaxe, collect the coal at the top of the level, return to the work table and furnace to build an iron pickaxe, with which it can clear the diamond blocking the goal.

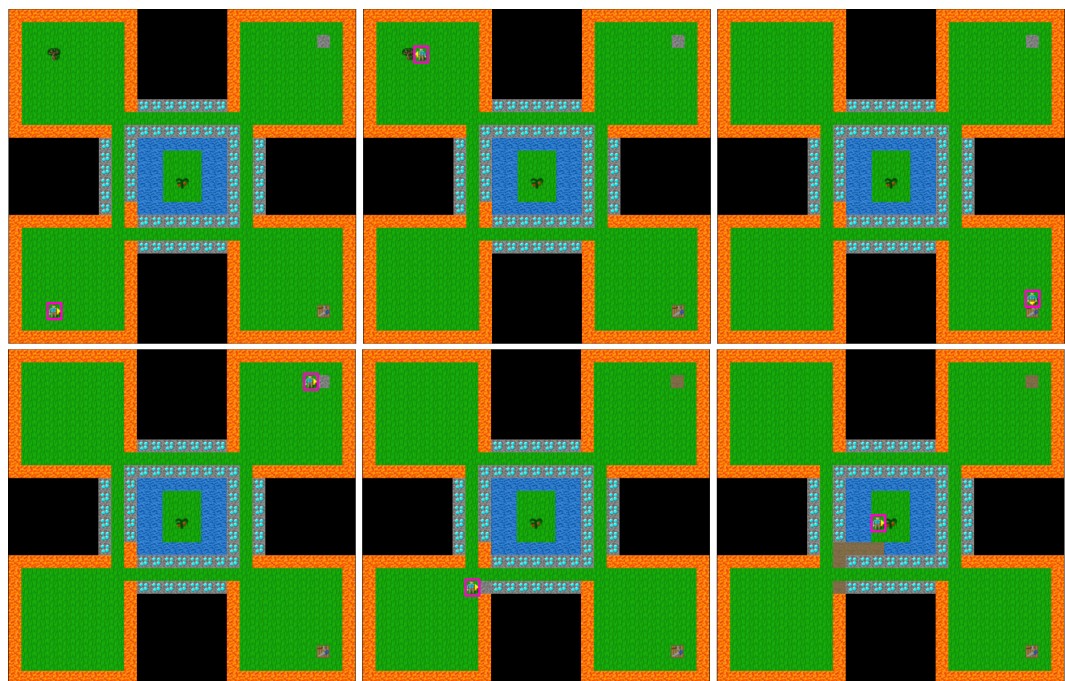

Figure 19: **Level 82.** The agent begins in the bottom-left corner and then must visit top-left corner to collect wood, visit the work table in the bottom-right to create a wood pickaxe, collect the stone in the top-right corner, and then successively place and remove the stone over the lava to create a walkable path through the lava corner in at the bottom-left of the central diamond-bordered square to reach the goal.

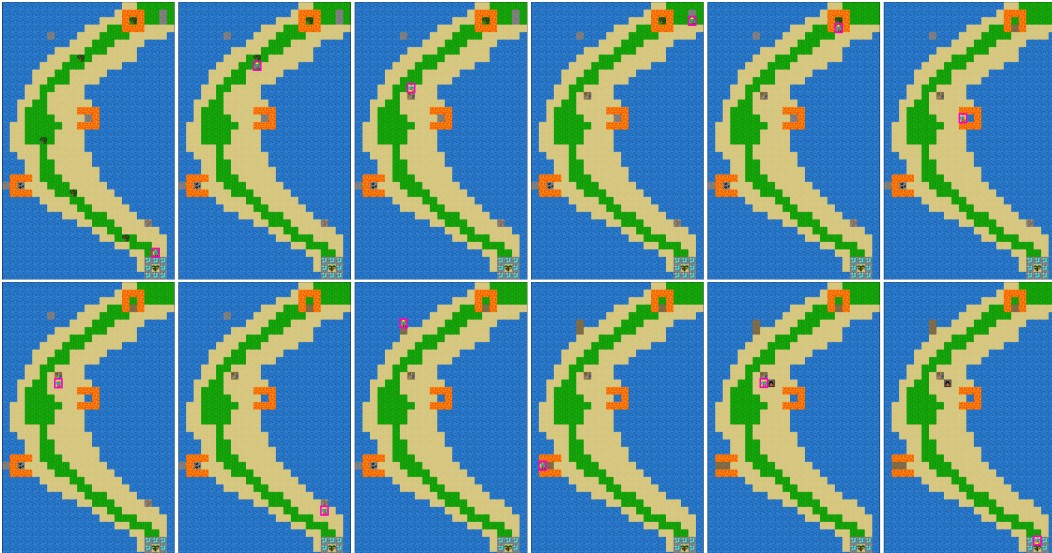

Figure 20: **Level 95.** The agent begins at the bottom of the island. It must collect the wood across the island to build a work table and then a wooden pickaxe, with which it can collect stone —which requires building a path to the stone in the water by placing and removing stone in the water. The agent must then return to the work table to build a stone pickaxe, with which it can collect the iron in the bottom-right. With the iron, the agent must return to the work table to create a stone pickaxe, which can be used to collect the coal, clearing the way to also place and remove stone over the lava to collect the final piece of stone. The agent must then return to the work table to build a furnace and then an iron pickaxe, with which it can use to clear the diamond blocking the goal.

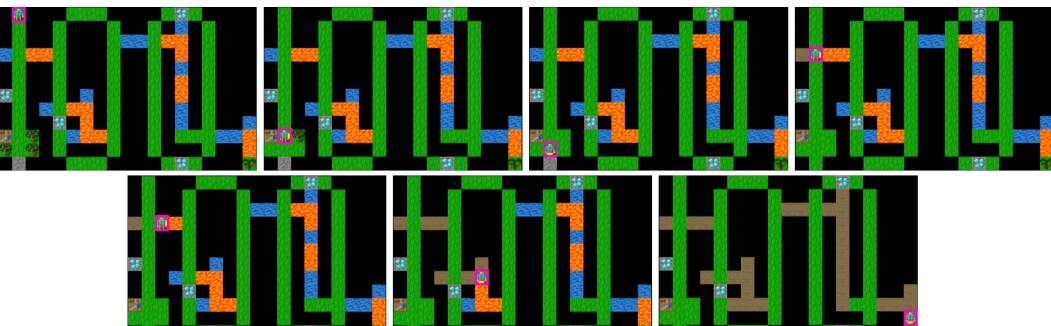

Figure 21: **Level 100.** The agent starts at the top-left, and must first move down to collect the wood and build a wooden pickaxe to collect the stone. With the stone, the agent must create walkable area over each of the crevices along the path in order to allow it to properly face the lava tiles, so that the agent can then place and remove the stone over the lava to create a walkable path towards the goal. This level creates difficulty by exploiting how moving towards water does not result in episode termination, while turning into lava does.

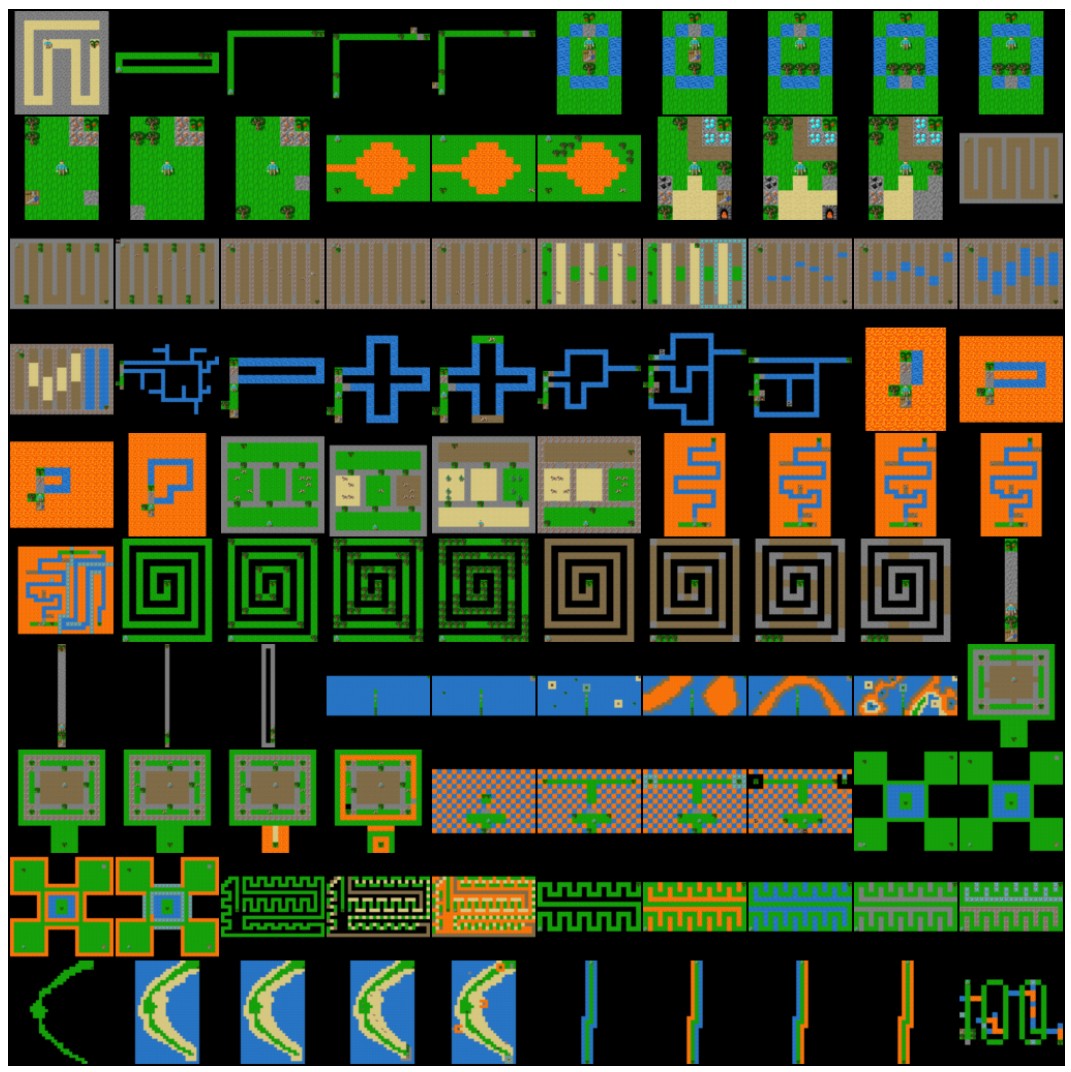

Figure 22: All 100 human-designed EscapeRoom levels, made using GriddlyJS.