# OpenReview forum: "GriddlyJS: A Web IDE for Reinforcement Learning"
_NeurIPS.cc/2022/Track/Datasets_and_Benchmarks — NeurIPS 2022 Datasets and Benchmarks _

### Official Review · Reviewer_dHJD · 2022-07-20
**A useful grid-world environment design tool via Web**

**Rating:** 7
**Confidence:** 3
**Correctness:** Most are correct to my knowledge.
**Clarity:** The paper is clearly written.

**Strengths:**

- Clear motivation, fully working system prototype and example.
- The web IDE provides a nice user interface, where I can easily create or edit a level of the provided environment with the visual editor.
- Backed by Griddly, the created environment can be easily transferred into Python for RL algorithm training.


**Weaknesses:**

- The system could be further optimized. E.g. I tried to add new level or delete the level, it takes quite a long time to process.
- To evaluate the agent's performance, we have to run the IDE locally and provide the model via ONNX format, which is not convenient.

Minor
- Typo around 748-749

**Additional Feedback:**

1. I notice that the example environment has a very long GDY file (over 5k lines). Maybe some of them are auto-generated (like the maps of different levels), but some need our efforts to code, right?
I can imagine this would be a very challenging and time-consuming task for environment designers.
Would you compare how much efforts needed using your IDE with other tools? E.g., I could also use the MiniHack Level Editor to graphically put objects on the map and the relevant codes are generated automatically.

2. Do you plan to provide some baseline results?

**Documentation:**

The documentation of Griddly can be found online.

**Ethics:**

No.

**Relation To Prior Work:**

Related works are clearly discussed.

**Summary And Contributions:**

This paper presents a web IDE, GriddlyJS, developed using React.js for reinforcement learning researchers to easily design and debug grid-world environments. It is based on the Griddly grid-world game engine. The tool also provides a user interface to visualize agent’s behavior, record trajectories and show the recorded replays. It allows users to design environments with procedural content generation, by designing environments of different levels, to evaluate agents’ robustness and generalization. The paper also includes a walkthrough of building such an environment as an example.

---

> ### Author Response · Authors · 2022-08-09
> **Thanks for the review!**
>
> Many thanks  for your helpful suggestions and comments! We’re glad that you see the many benefits provided by GriddlyJS to the RL research community. We now aim to address your concerns below:
>
> ### Reducing latency in the user interface
> The Grafter Escape Rooms environment features complex transition logic and 100 individual levels. This complexity contributes to the latency in the user interface when interacting with this environment. We plan to track the latency issues you mention and will aim to further optimize the latency for future versions of GriddlyJS. We have created a GitHub issue to track these improvements here: https://github.com/Bam4d/Griddly/issues/231. Note that these latency issues do not appear in less complex environments.
>
> ### Running the IDE and models locally
> We agree that the experience of evaluating agent performance would be much improved with the addition of a  fully persistent data store to the hosted version of GriddlyJS. However, this requires implementing a dedicated web server allowing for saving, uploading, and persisting models—a significant engineering challenge requiring dedicated financial support (for hosting). In the meantime, we have added further details on how to convert models to TF.js for evaluating agents in GriddlyJS locally here in Appendix B.4, linking to our tutorial here: https://github.com/GriddlyAI/escape-rooms#using-checkpoints-in-griddlyjs
>
> ### Additional Feedback
> We agree that the GDY file format can be quite verbose in terms of the number of lines of code, but the majority of these lines, as you mention, can be automatically generated (e.g. by providing additional logic and common blue-print behaviors to reduce boilerplate code). The actual behavioral description of the environment is approximately 1.9k lines. Because these environments are running in a C++ framework with GPU rendering acceleration, 1.9k lines is actually relatively small in comparison for implementing an equivalent, performant environment from scratch in these languages. Moreover, there is no need to think about optimization of these environments as this is done by our framework.
>
>
> ### Providing baseline results
> We provide baseline results for the 100 escape-room levels generated using GriddlyJS and compare it against a strong agent trained using Domain Randomization (Figure 6). Our results show that these human-designed levels are very challenging (much more so than the randomized levels) and believe they are a good example of the datasets that can be created using GriddlyJS. We believe such datasets (including the Grafter Escape Room levels we provide) open many additional avenues of research, exploring the interplay between humans and machine learning algorithms in designing levels for training robust RL agents.

---

### Official Review · Reviewer_5bee · 2022-07-24
**Handy and useful web-based IDE for Gridworld RL environment development**

**Rating:** 8
**Confidence:** 3
**Clarity:** The paper is well-written and easy to…

**Strengths:**

1. The introduced web-based IDE, that can be run in any web browsers with visual interface, is high-quality and easy to use, so it would be helpful for from beginners to experts in RL society.
2. Both the documentation website and Github repository are well-organized with good examples.


**Weaknesses:**

1. The experiment is conducted only on a single environment (Figure 6). Further analysis on various environments would be helpful to highlight the difference between DR and human-designed levels.
2. The authors expect that human-in-the-loop environment design would mitigate out-of-distribution problems and make RL agents robust, but no experimental result has provided. If some experimental results are provided at least on a toy domain (as following the manner described in line 280), it would emphasize the usefulness of human-in-the-loop level design.
3. As described in line 342, GriddlyJS does not currently persist user-generated data on a dedicated server, but I think it is very important to guarantee that GriddlyJS supports active sharing among RL researchers.


**Additional Feedback:**

- Question
    * Is it available RL agent with codebase written in PyTorch framework on GriddlyJS? Since both of TensorFlow and PyTorch framework are widely used by RL researchers, support (maybe in the future) for both frameworks would be very helpful.

- Minor typo
    - In section 4.3, line 294, “wide range of downstream use cases.” (missing “.”)


**Correctness:**

Most of claims sound correct, but more experiments on the usefulness of human-in-the-loop level design would be helpful as in second paragraph of weakness.

**Documentation:**

The documentation provides the sufficient details.

**Ethics:**

No.

**Relation To Prior Work:**

The authors clearly discussed how this work differs from previous works in section 5.

**Summary And Contributions:**

This paper presents a web-based integrated development environment (IDE), GriddlyJS, especially for Gridworld reinforcement learning (RL) environments. It abstracts the details of implementing RL environment that requires skillsets of game developers and make RL researchers quickly design new environments by themselves even without such skillsets. By experimental comparison between procedurally generated levels and human-designed levels, it is demonstrated that human-in-the-loop level design is helpful to make RL agents robust. Furthermore, GriddlyJS streamlines the RL research procedure consisting of environment development, agent evaluation, and publication so that empowers research productivity and reproducibility in RL society.

---

> ### Author Response · Authors · 2022-08-09
> **Thanks for the review!**
>
> Thank you for taking the time to review our work. We’re really glad you like the paper! We generally agree with your feedback and seek address them below:
>
> 1. We agree that similar experiments on a wider range of PCG environments will add further support for the hypothesis that human-designed levels can be more challenging than randomly-generated ones. However, given the stark contrast between performance on human-designed levels and randomized levels, it seems the Grafter results strongly support this hypothesis. In general, humans can directly adversarially design environment configurations found to be particularly challenging to the pre-trained policy. GriddlyJS’s human-in-the-loop level editor allows this fact to be directly exploited in generating a wide range of challenging scenarios for agents, and we believe this will make for many interesting future works built on top of GriddlyJS.
>
> 2. It would be possible to train an agent directly on the human designed levels or by combining these human levels with automatically generated levels produced by automatic curriculum learning methods such as ACCEL or PLR. We believe such additional experiments can be considered interesting follow up work—which is made significantly easier by GriddlyJS.
>
> 3. We agree here that the data being stored locally is not ideal, but to persist this data in an online account requires a significant engineering effort, as well as financial support. We plan on supporting this in a future release of GriddlyJS. We did not want to wait for support for this additional feature to block the release of GriddlyJS in its current form, whose core features can offer significant benefits to the research community.
>
> Regarding your question about using TF.js models: There is freely available tooling to convert models from many frameworks to be compatible with TF.js, including PyTorch and JAX. For example, conversion from PyTorch to TF.js via ONNX: https://learnopencv.com/pytorch-to-tensorflow-model-conversion/
> Our experiments in Figure 6 are trained using PyTorch and converted to TF.js using a similar method, and this model is uploaded to GriddlyJS. We have now included these details in Appendix B.4, including a tutorial on how to convert models to TF.js for loading into GriddlyJS with a helper script, also included in the GriddlyJS repository: https://github.com/GriddlyAI/escape-rooms#using-checkpoints-in-griddlyjs
>
> Thank you again for your positive assessment of our work. We hope this additional discussion addressed the questions and comments you raised in your review.

---

> > ### Comment · Reviewer_5bee · 2022-08-22
> > **Response**
> >
> > I thank the authors for addressing the concerns. Still, I hope that the authors will support persistent server maintenance for user-generated data or environments in future. If so, I expect GriddlyJS can be very useful software for RL researchers.

---

### Official Review · Reviewer_Gxpm · 2022-07-27
**The paper presents a useful wrapper for the web around the `Griddly` framework to develop grid-based environments, but lacks the possibility to properly evaluate RL agents.**

**Rating:** 6
**Confidence:** 4

**Strengths:**

- Lowers the barrier to create novel grid-world environments.
- Grid-world environments are a good testbed for RL as they present a moderate complexity that can easily be tuned and become more complex.
- Procedural content generation in `Griddly` enables the definition of interesting RL environments with test and training distributions which can easily be tried and shared. Though this is rather a property of `Griddly` and not its web wrapper.

**Weaknesses:**

- Grid-world environments are only relevant to certain kinds of RL researchers, which exclude continuous control or environments with more complicated dynamics. However, this limitation does not speak against its acceptance.
- Debugging of a policy online does only work for the `Grafter Escape Rooms`  environment. Locally this works only with ONNX exportable models. This is the major weakness for a tool to test RL agents.
- Automatic performance evaluation of the policy on a set of levels would also be nice to have and aid reproducibility

**Additional Feedback:**

- Link in the help to the model directory (https://github.com/Bam4d/Griddly/tree/level_editor/js/griddlyjs-app/public/model) is broken.
- In Firefox on MacOS, the user has to scroll to see the whole UI. Its elements should be responsive to prevent this.

**Clarity:**

- The paper is well written and has a good structure.

**Correctness:**

- The tool is presented in a sound way, though a major drawback is that evaluating an RL agent does not work in a simple way. This is not apparent from the paper.

**Documentation:**

- The help dialog is useful though there are several typos in it which are mostly whitespace errors.

**Ethics:**

- I have no ethical concerns.

**Relation To Prior Work:**

- The paper gives a nice overview over the different approaches to procedurally generated grid wolds.

**Summary And Contributions:**

- The paper introduces a WebAssembly-based wrapper around the `Griddly` framework for grid-world environments.
- The tool enables the definition, play and recording of trajectories of grid-based RL environments directly in the browser.

---

> ### Author Response · Authors · 2022-08-09
> **Thanks for the review!**
>
> Many thanks to the reviewer for their useful comments and suggestions, which we believe have improved our submission. We are glad you found our work to lower the barrier to developing novel RL environments and to provide a good testbed for RL methods. We now seek to address your main concerns, and hope that based on this further discussion, you will consider increasing your rating of this work:
>
> ### Support is limited to ONNX-compatible models
> Thank you for this suggestion. We fully agree that a general online evaluation platform for RL policies would be valuable (similar to how Dynabench https://dynabench.org/ has been valuable for the NLP community). However, we believe GriddlyJS, in its current form, already provides significant value to the RL community. While the current version requires ONNX-compatible policy models, the most common frameworks all support export to ONNX, including PyTorch, JAX, and TensorFlow. GriddlyJS ultimately supports the evaluation of any policy model that can be exported to TF.js to run in the browser.
>
> To simplify the process of converting a researcher’s policy model to the necessary TF.js format, we include a script in the GriddlyJS repository for easily converting Pytorch models to TF.js: https://github.com/GriddlyAI/escape-rooms#using-checkpoints-in-griddlyjs. We have updated the manuscript to highlight these points (See appendix B.4).
>
> We have only uploaded the model for the Grafter Escape Rooms project as an initial demo. This does not stop other users from hosting their own (locally or publically) models for different Griddly environments. The open source components provided in GriddlyJS can be modified to run on many web-based mediums such as projects websites, github pages, or blog posts.
>
> ### Missing automated policy evaluation over multiple environment configurations
> Thanks for suggesting this feature. We agree that adding the ability to automatically evaluate and reporting performance of an imported TF.js policy over multiple environments would be a useful addition to GriddlyJS. We have created a Github issue tracking development of this feature here: https://github.com/Bam4d/Griddly/issues/228. While GriddlyJS does not currently support this sort of batch evaluation, it does allow researchers to easily visualize any policy’s performance on any confirguration of the environment, which provides significant value to researchers in simplifying policy debugging and analysis, as well as in sharing reproducible results.
>
> Thanks also for pointing out user interface issues while providing specifics such as the browser and OS version, We have included a github issue to highlight this in general: https://github.com/Bam4d/Griddly/issues/230.
>
> Given this additional discussion, we hope you will consider increasing your support for our work or otherwise describe what stands in the way for your doing so.

---

> > ### Comment · Reviewer_Gxpm · 2022-08-13
> > **Response**
> >
> > I want to thank the authors for addressing my main concerns and am looking forward to see the changes implemented in the future.
> >
> > After reading the other reviews, I see the main merit of this work to be the collection of offline RL data and the easy visualisation of games and agents via a web-based interface. Thus, even though main concerns are not fixed yet, I will raise my rating in good faith that the acceptance will lead to an increased effort to improve `GriddlyJS`.

---

> > > ### Author Response · Authors · 2022-08-19
> > > **Thanks for your support!**
> > >
> > > Thank you for your prompt response and your belief in GriddlyJS!
> > >
> > >
> > > We will prioritise these features as we agree that they will be very useful for the community.

---

### Official Review · Reviewer_qU5s · 2022-07-27
**Useful Software, but is it a Benchmark?**

**Rating:** 7
**Confidence:** 2
**Clarity:** The paper is well-written and easy to…

**Strengths:**

The paper highlights the importance of procedurally generated content (PCG) in reinforcement learning research. In particular, the paper outlines why PCG is important for generalizable and robust RL algorithms. The paper also provides strong evidence for why gridworlds are an important tool for testing the generalizability of RL algorithms, which is useful to support the need for GriddlyJS.

GriddlyJS turns the Griddly into a somewhat no-code framework, which can increase the accessibility of creating RL environments.  Users who have little coding experience can more easily create content for RL research.


**Weaknesses:**

The level to which GriddlyJS is a contribution is unclear.

The main argument the paper presents is that environment construction for RL research is difficult since it requires the skills akin to video-game development. To that end, the paper presents GriddlyJS, which can help mitigate the need for these skills.  Although I am not familiar with Griddly, it seems like Griddly is a framework that already mitigates the needs for video-game developer skills.  How does GriddlyJS make working with PCG content easier than Griddly does, besides using a GUI? It seems as if GriddlyJS is simply a graphical extension/update to Griddly.

Ideally, a new benchmark would provide researchers with a new way to characterize or test algorithms for which existing benchmarks cannot be used. This does not seem to be the case with GriddlyJS.

The experiment on PPO was not described in detail. How were hyperparameters chosen? How many runs were used? Furthermore, I am unsure of how the experiment differentiates how GriddlyJS can show the out of distribution generalization of PPO better than Griddly can since, from my understanding, environments can also be hand-designed with Griddly (just not using a GUI).

Overall, GriddlyJS seems more like a software engineering project than a research contribution. GriddlyJS may be better suited to the JMLR open-source ML software track than to the Neurips benchmark track.


**Additional Feedback:**

The following issues did not affect the scoring of the paper:

Line 143: "GDY" has already been defined above.

Line 294: Missing period at sentence end

**Correctness:**

The claims made in the paper are mostly correct. As I previously stated, I am not very familiar with Griddly, but it seems as if some of the characteristics attributed to GriddlyJS should actually be attributed to the Griddly backend.


**Documentation:**

GriddlyJS provides sufficient documentation on how to use the framework.  In particular, the framework works seamlessly with Griddly, which is heavily documented.


**Ethics:**

No ethical concerns exist with this work.

**Relation To Prior Work:**

The paper thoroughly discusses the relation of GriddlyJS to other PCG environments for RL research. The paper highlights how GriddlyJS differs from these other benchmarks.


**Summary And Contributions:**

This paper introduces a new Web intelligent development environment (IDE) for creating gridworld environments, GriddlyJS. GriddlyJS is an easy-to-use IDE for creating procedurally generated gridworlds, where the environment designer can edit and modify these gridworlds using code or a GUI. Users can also create a set of hand-designed gridworlds for reinforcement learning algorithms to learn on. GriddlyJS is different from Griddly in that it is a frontend for accessing Griddly in an easier way.

---

> ### Author Response · Authors · 2022-08-09
> **Thanks For the Review!**
>
> Thank you for taking the time to review our work and for your suggestions and comments. We are glad you see the benefits of the no-code approach offered by GriddlyJS. We now seek to address your concerns, and hope that in light of this discussion, you will consider increasing your support for our work.
>
> ### Is GriddlyJS simply a software engineering project?
> We respectfully but firmly disagree with your perception that the GriddlyJS tool is simply a software engineering project. We believe GriddlyJS holds significant academic merit in advancing what is possible for the average researcher working in RL: GriddlyJS unlocks many avenues of research that would otherwise not be possible. Firstly, the level editor component of GriddlyJS makes the creation of complex and diverse human-generated levels possible without requiring the user to understand the underlying game engine or have any knowledge of programming. The benefits of this are manyfold:
> * First, Griddly significantly reduces the time required for researchers to build challenging environments for their experiments, accelerating the pace of RL research.
> * Second, it allows the design of new levels or variations of these environments to be offloaded via an interactive web interface, possibly to the research community or the public, allowing massive datasets of diverse level variations and agent performance therewithin to be crowdsourced. Streamlining human level design in this way is powerful, as existing procedural content-generation algorithms are in their infancy and do not necessarily match the creativity of human designers in finding interesting and challenging scenarios (for both humans and AI agents). Such human-generated levels serve as both useful, adversarial training levels for RL agents, as well as a valuable dataset for improving PCG algorithms. Figure 6 highlights the uniquely challenging properties of human-designed levels over randomly generated ones.
> * GriddlyJS also makes it easy to record human demonstrations in these environments, thereby serving as a web-based interface to collect massive amounts of training data for offline RL, a rapidly growing area of research.
> * GriddlyJS also makes it trivial to visualize the behavior of any policy inside the an interactive, visual environment, allowing researchers to better understand the strengths and weaknesses of their models and training algorithms.
> * Moreover, GriddlyJS makes it significantly easier to share reproducible results in the form of interactive environment-agent demos on the web, greatly promoting reproducible research.
>
> Additionally, we would like to point out that for the Datasets & Benchmarks track, “Data generators, reinforcement learning environments, or benchmarking tools are also in scope.” GriddlyJS is a useful human-in-the-loop generator of gridworld environments, which ultimately provides a powerful tool for quickly generating an endless variety of new benchmarks.
>
> Regarding the PPO hyperparameters, we have now added significantly more experiment details in Appendix C.
>
> Given this further discussion, we hope you will consider raising your rating of our work to “Accept,” and if not, that you will describe what stands in the way to a higher rating.

---

> > ### Comment · Reviewer_qU5s · 2022-08-26
> > **Change of Rating**
> >
> > I thank the authors for the clarifications made. After considering these clarifications as well as the discussions between the authors and other reviewers, I have decided to increase my rating of the paper.
> >
> > Although, from my understanding, many of the features in GriddlyJS are already available in Griddly, the GUI introduced by GriddlyJS does indeed increase the accessibility of the Griddly framework. In addition, the GriddlyJS framework introduces a few new features that will be helpful to researchers.
> >
> > I also thank the authors for the additional experimental descriptions included in the revised version of the paper.

---

### Official Review · Reviewer_yTnK · 2022-07-28
**An intuitive and useful tool for RL research which needs some more documentation**

**Rating:** 7
**Confidence:** 4

**Strengths:**

I believe a tool like GriddlyJS could greatly impact the research field of RL by (1) simplifying the process of designing new tasks, (2) understanding tasks by interacting with them, (3) simplifying analysis of trained policies with its step-by-step rendering tool in modifiable tasks, and (4) allowing to publish informative demonstrations of agents. This makes for great potential value and significance of GriddlyJS and is also novel to the best of my knowledge. I believe the design, manipulation and analysis of existing environments is likely the most impactful and important aspect of GriddlyJS.

The UI of GriddlyJS is intuitive and simple with most functionality easily accessible.


**Weaknesses:**

1. In line with its potential and rich feature set, it appears essential to provide detailed documentation and a simple process for its features. However, I find the current documentation despite its simple UI lacking with several features mentioned in this work missing descriptions. See two main unclear components below with less pressing issues listed under the documentation section of this review.

    1.1. TF.js models: The publication and visualization of agent policies for reproducibility, debugging, and analysis is presented as a core feature of GriddlyJS. However, it is not explained how TF.js models can be integrated within GriddlyJS.

    1.2. Multi-agent support: Griddly supports multi-agent environment and this work also mentions support for multiple agents. How can these multiple agents be controlled in the human “play” interface? It appears only the first agent is controlled with the assigned keybindings and other agents stay static while episodes go on as long as any, even static, agents are still alive (tested in Crafter and Spider environment with manually multiple agent entities being placed in the environment).

2. The evaluation presented in Figure 6 appears confusing to me.

    2.1. It appears that the randomly generated DR levels are already solved with ~50% before any training and no progress is apparent throughout the entire training. This suggests that 50% of the DR generated tasks are trivial (highlighting the relevance of human-generated tasks) but also indicates that no significant training is exhibited throughout the entirety of training rendering the learning curve invaluable. If I correctly interpret these results, the claim might also be that a random policy solves 50% of DR-generated levels while human levels are only solved with a significantly smaller probability.

    2.2. The PPO agent only seems to be trained for 300 updates which appears like very little training. Is it correct that each update is computed with batches of 512 steps (PPO rollout length) x 64 (64 PPO workers) for a total of ~10M timesteps?

**Additional Feedback:**

I believe the usefulness and impact of GriddlyJS largely depends on its ease of use to lower the barrier of entry. In line with this thinking, I would like to propose two features which I believe could greatly simplify the use of GriddlyJS and improve its adoption:
1. It would be useful to provide an option to directly download created environments as gym environment (e.g. as .py file which registers a gym environment) and collection of environments within a project. It appears that GDY files could be copied into a file and loaded with Griddly but automating this process could simplify this process significantly and improve GriddlyJS’s value as an IDE for RL environments.
2. Designing new environments rather than new tasks within pre-existing projects within GriddlyJS still requires some expertise with GDY. Its interface is fairly intuitive and I commend its authors for its design, but it might be possible to simplify the generation of new environments by providing blueprints/ prototypes for objects with common features. Example could be collectible objects and locations which give reward (positive or negative) and potentially terminate episodes, movable or static enemies and similar generic concepts. These could be loaded and easily modified but serve as a starting point for environment designers to use. The Crafter environment is a complicated example with large amounts of logic, but its GDY files shown in GriddlyJS has about 2400 lines to specify all objects, logic, their appearance etc. Designing such environments currently still requires significant expertise with Griddly and would likely take large amounts of efforts and time.

**Clarity:**

The paper is clearly written. I found a single typo in l. 182 (“... added to the it.” --> “... added to the grid.”).

**Correctness:**

To the best of my knowledge, the claims in the paper appear correct. I have some concerns and questions regarding the evaluation presented in Section 4.2 - see weakness 2. for more details.

**Documentation:**

I believe the clarity and usability of the GriddlyJS web-tool is essential to evaluate the merit of this work. I tested the GriddlyJS platform on Windows 10 with the Google chrome browser. It would be great if more extensive documentation/ explanations could be provided within GriddlyJS to explain its functionality deeper. For two pressing components, see the respective comment under weakness 1. Furthermore …
1. The level editor does not seem to allow scrolling through the list objects which is relevant for environments like Grafter with many objects.
2. The paper refers to varying observers of Griddly to render environments in different ways with the 2D observer apparently being used by default in GriddlyJS. Is it possible to use other observers in the web IDE as well?
3. How can assets be added and/ or selected within GriddlyJS to render defined objects?

**Ethics:**

The work does a good job of illustrating the broader (ethical) impact of this work in Appendix F. I do not see any ethical concerns that warrant further discussion.

**Relation To Prior Work:**

The work does a good job of explaining the underlying Griddly engine and positioning the work within related systems.

**Summary And Contributions:**

This work introduces the GriddlyJS platform, a web-based IDE for RL based on the previously introduced Griddly engine. The web-tool provides functionality to design, test, and analyse RL environments based on configuration files for the Griddly engine. The tool also provides functionality to visualise, evaluate, and record human and agent episodes which can be used to analyse trained agents, better understand tasks, and publish interactive policy demonstrations. As an example, an online version of the Crafter environment is implemented and released in GriddlyJS, and an experiment demonstrates that manually defined tasks in the new environment are significantly more challenging than procedurally-generated, random environments.

---

> ### Author Response · Authors · 2022-08-09
> **Thanks for the review!**
>
> Thank you for your comments and suggestions! We’re really glad you see the main benefits of GriddlyJS in streamlining the development and analysis of reinforcement learning environments.
>
> ### Running TF.js models in GriddlyJS
> TF.js is the de-facto method of running neural networks in the browser, but it does not restrict the training of networks in any way. There is a simple conversion process between any other training frameworks which uses the ONNX format, which is universal across many existing neural networks libraries. Our experiments in this paper are actually performed using PyTorch and converted to TF.js.
>
> We have now clarified this process and link to documentation detailing this process in the updated manuscript, as well as the help section of the GriddlyJS user interface. Specifically, we include references to this tutorial on loading models in GriddlyJS: https://github.com/GriddlyAI/escape-rooms#using-checkpoints-in-griddlyjs
>
>
> ### Multi-agent support
> Griddly itself allows multi-agent and multi-agent and multi-unit RTS style games, however including these in GriddlyJS would require significant engineering effort. We plan to include this in later versions of GriddlyJS. (As GriddlyJS is an open-source project, we endeavour to continuously improve and add features.)  The reason multiple-agents are allowed in the current version (though only one is controllable by the policy) is so that we don’t actively restrict development of multi-agent environments. This design decision keeps the door open for full multi-agent support in GriddlyJS.
>
> Mechanics between agents can still be tested (from the perspective of a single agent). We’ve added a ticket to improve this in future versions https://github.com/Bam4d/Griddly/issues/223
>
> ### PPO Evaluation
> We have re-run this experiment as we had similar concerns to the reviewer in this respect. We have discovered and fixed several issues we had in our implementation, and the updated results are now presented in Figure 6. We have also extended Appendix C with comprehensive experimental details.
>
> Thank you also for your comments regarding certain features and documentation of GriddlyJS. We have created GitHub issues tracking the improvement of each of these details you point out here:
> * Item scrolling: https://github.com/Bam4d/Griddly/issues/224
> * Renderer selection: https://github.com/Bam4d/Griddly/issues/225
> * Adding assets: https://github.com/Bam4d/Griddly/issues/226
>
> The features you describe in “additional feedback” are great suggestions for improving GriddlyJS! We wholeheartedly agree with these suggestions. The automatic generation of a Gym-compatible `.py` file is a fantastic idea, and we would like to add this to our roadmap. Further, supporting blue-print behaviors in GDY, which can be further configured in a simple manner to avoid a large amount of boilerplate code, is a feature we have also considered and hope to include in GriddlyJS in a future release. We have added a github issue to track this feature: https://github.com/Bam4d/Griddly/issues/234
>
>
> Given this additional discussion and related improvements to our paper and documentation, we hope you will consider increasing your rating of our work to “Accept.”

---

> > ### Comment · Reviewer_yTnK · 2022-08-12
> > **Response to Rebuttal**
> >
> > I thank the authors for their rebuttals, provided clarifications, and improvements made to their submission. I have read all other reviews and rebuttals and still believe that this work will benefit the RL research community positively.
> >
> > ### TF.js models
> > I appreciate the clarifications and added documentation. I agree with the comments of reviewer dHJD that this functionality could be further improved, but am convinced that the ability to evaluate TF.js models online will be useful to researchers (independent of their DL library of choice).
> >
> > ### Multi-agent support
> > Seeing such support in the future would be a great addition given the quickly growing community researching multi-agent reinforcement learning.
> >
> > ### PPO Evaluation
> > I thank the authors for their efforts improving the evaluation. This part now looks much more clear to me. Some remaining questions/ concerns I would have about the evaluation:
> > - The authors do not specify which DR levels they evaluated the policy in. Presumably, these are levels generated within the same "EscapeRoom" scenario used during training. In this case, it is unclear to me whether the result indicates that human levels are more interesting/ challenging or whether these are merely "out-of-distribution" for the trained PPO agent whereas the DR levels would be "in-distribution" generalisation which we would expect the agent to be able to achieve after sufficient training. Is this a fair comparison? How "similar" or "different" do the authors consider the set of procedurally generated environments and the set of manually designed tasks?
> > - Also, the claim is made that "the resulting policy solves DR levels at a significantly higher rate than human-designed levels, highlighting the distinct quality of human-designed levels". However, the same performance as achieved in human-designed levels could be observed when evaluating in randomly generated tasks as long as these are sampled from a distribution of sufficiently difficult tasks. While I still agree with the authors' intuition that procedurally generated levels can often fail to capture diverse and interesting challenges across tasks, and I would hypothesise humans with some domain knowledge to create such challenging levels comparably easily, the provided evaluation does not provide absolutely convincing data for these claims.
> >
> > I would be happy to further engage in discussion should the authors have any further clarifications or thoughts on this matter!

---

> > > ### Author Response · Authors · 2022-08-19
> > > **Further Discussion**
> > >
> > > Thanks for your prompt response and continued support of the GriddlyJS framework!
> > >
> > > ## PPO Evaluation
> > >
> > > Thanks to the reviewer for pointing out this discussion point in our paper.
> > >
> > > We have added a Domain Randomization (DR) section to Appendix C.3.2 which highlights how the levels are generated to make this much clearer to the future readers. We describe our modifications to the Crafter environment’s level generator and include examples of the generated levels.
> > >
> > > We also note that the DR levels represent a distinct distribution to the levels generated by humans. Generating levels that are both challenging and diverse in required skillsets—such as those designed by human game developers—is ultimately a search problem in a high-dimensional design space. Finding specific solutions through randomly sampling the design parameters, such as those represented by human-designed levels, would require exponential sample complexity in the number of choices per design parameter. This highlights the importance of using human-levels. Such differences between random levels (which are a kind of noise) vs. the often highly-structured human-designed levels explains why agents trained on DR are unable to succeed on the more structured human levels.
> > >
> > > We have also added a paragraph L304-308 with several references supporting our claims that generation of challenging and diverse levels resembling human-designed levels is out of reach of current PCG methods, which are largely based on randomization approaches.

---

> > > > ### Comment · Reviewer_yTnK · 2022-08-23
> > > > **Further Discussion**
> > > >
> > > > Apologies for the late response. I thank the authors for providing further insight into the PPO evaluation and decided to increase my score.

---

### Official Review · Reviewer_pM8Q · 2022-07-28
**A useful and practical IDE  for RL, but requires better justifications to put it in better context  and/or broader scope**

**Rating:** 6
**Confidence:** 3
**Correctness:** N/A

**Strengths:**


i) An IDE towards generating an environment and evaluating agents, which is of practical use
ii) Ability to collect human trajectories

**Weaknesses:**

i) Only supports Grid Environments. In section 2.3 they make a good case for why the grid world is important. However, I think to improve the manuscript it should be discussed that, what RL aspects the paper/IDE currently cannot cover due to this focus on the grid environments. Also, see the related following point.

ii) I understand the "portability" a web IDE gives, or how it can work as a central point, however, does the choice of Web IDE limits the supported environments? a) How much effort would be needed to support environments like strategy games like StarCraft [1], Minecraft, RObotics, or multi-agent sports environments like Soccer [2], which might require much more "graphical" capabilities than grid world environments? b) Why a Web interface is preferable over A traditional Python package/ software? I think this should be discussed in the paper and also included in the introduction at least briefly.

[1] Vinyals, Oriol, et al. "Starcraft ii: A new challenge for reinforcement learning." arXiv preprint arXiv:1708.04782 (2017)

[2] Kurach, Karol, et al. "Google research football: A novel reinforcement learning environment." Proceedings of the AAAI Conference on Artificial Intelligence. Vol. 34. No. 04. 2020.

iii) A human study on hinting how it can help different aspects of an RL researchers productivity would be a very strong addition to the paper.

**Additional Feedback:**

1)
Line 32-35
" developing new environments that provide the necessary challenges for RL methods
 is a costly process, requiring deep expertise in software engineering, high performance computing
(for efficient distributed training), and game design.  ...This skillset is closer to that of a videogame
35 developer than a typical machine learning researcher. "

--- the way it is written, it seems to be claiming that PCG algorithms can already generate game levels that are close to that are generated by Video game developers and better than what SOftware developers can generate. I would recommend to Either substantiate  this with references or rephrase to avoid accidentally making claims that are not really in scope of the paper.

2) Line 38-39 " i.e. creating probabilistic programs
39 that specify distributions over environment configurations, adds considerable engineering overhead to". Please add references, recently efforts have been made to make environment generation easier using Probabilistic Languages for gridworld or real time games providing compilers and helper libraries.




**Clarity:**



Mostly the paper is well written.
I have one point:

i) Does it Only allow agents that are written in Tensorflow and/or converted to TensorflowJS ?. I am not entirely sure about this point. Can PyTorch models be used too?


A few other clarity points are discussed in the other sections of this feedback.

**Documentation:**

The website is good. However as the focus is on the interface, the initial help screen's navigation can be made better. I was expecting a simple workout grid with the corresponding DSL on the welcome screen. Some interactive help on the DSL would be a great bonus.

All the above comments, also acknowledge that a web interface can be improved forever and the current version is good enough as the first  version.

**Relation To Prior Work:**


1) Previous works have also focused on using Probabilistic programming languages/DSLs with similar motivations such as generalization, ease of environment generation, and compilation of the environments abstracting away lower level details. I think the manuscript can be improved by adding a small section making a comparison to other similar works e.g., PCG/DSL for Soccer[1], NetHack[2] (which is mentioned in the paper) on i) The choice of the DSL and  How it differs, ii) on the level of abstraction, iii) mode of use iv) user friendliness/capabilities of the DSLs.

[1] Azad, Abdus Salam, et al. "Programmatic Modeling and Generation of Real-Time Strategic Soccer Environments for Reinforcement Learning." Proceedings of the AAAI Conference on Artificial Intelligence. Vol. 36. No. 6. 2022.

[2] Samvelyan, Mikayel, et al. "Minihack the planet: A sandbox for open-ended reinforcement learning research." arXiv preprint arXiv:2109.13202 (2021).


**Summary And Contributions:**

The authors Proposed A Web IDE  to

i) decrease the complexity of environment generation using PCG
ii) simplify the build process
iii) reproducibility

which allows:
i) design and debug arbitrarily, complex PCG grid-world environments
ii)  visualize, evaluate, and record the performance of trained agent models in the Web


The work is a useful one. However, I think the justifications can be heavily improved to put this work in a better context with the current related works. Also, It is unclear whether it is possible to incorporate other environments and if possible how much engineering effort would be needed. Even in the case, that further incorporation of non-grid-world environments is not possible, the manuscript should outline what would be the implications of that. I am looking forward to the rebuttal and am open to changing my rating.

---

> ### Author Response · Authors · 2022-08-09
> **Thanks for the Review!**
>
> Thank you for your useful feedback, which has helped us improve our work. Further, we are glad that you find GriddlyJS a useful tool for RL researchers. We now seek to address your main concerns around our work:
>
> ### Missing discussion on the limitations of gridworlds
> Many thanks for pointing out that, while we discuss the benefits of gridworlds for RL research, we should also discuss their limitations. We have now updated the manuscript with a more detailed discussion of the limitations of gridworlds in Section 2,3.
>
>
> ### Does GriddlyJS allow arbitrary RL environments to run on the web?
> Providing support for running arbitrary RL environments, e.g. StarCraft, on the web is not the goal of GriddlyJS. Our work specifically proposes the gridworld format as an intuitive domain for researchers and even non-technical persons to generate new level designs for RL agents, whilst also preserving the ability to benchmark many of the major challenges for RL agents. Notably, based on these motivations, one of the highest profile competitions in NeurIPS 2021—the NetHack Challenge—centered on a gridworld. GriddlyJS significantly lowers the barriers for developing new gridworld environment benchmarks, and reduces the costs of debugging and analyzing policies in these environments. Importantly, GriddlyJS provides a web-based interface for sharing the ensuing experiments and results with the greater research community, thereby promoting reproducible research.
>
> ### Why is GriddlyJS provided as a web interface rather than a traditional python package?
> GriddlyJS is designed as a web interface, because this removes the need for users to build or install any additional software. Crucially, as a fully web-based app, GriddlyJS makes it possible to directly share new environments and interactive agent-environment demos directly on the web, thereby promoting reproducible RL research as well as the gradual sharing and compounding of useful research artifacts. In future releases of GriddlyJS, we plan to make it possible to directly embed an environment-agent demo generated in GriddlyJS inside a separate webpage, e.g. a blog post or demo website. These specific advantages of designing GriddlyJS as a web-based interface are outline in our paper in the Abstract (L12-15) as well as the Introduction (L63-71).
>
> ### Running a quantitative human study
> The reviewer also suggest a quantitative human study on how GriddlyJS improves the productivity of researchers. We believe that such quantitative studies of user productivity gains using IDEs is generally interesting to consider for future work in human-computer interaction. Now that GriddlyJS framework is publicly available, this suggestion highlights one of many possibilities of different research directions that GriddlyJS enables other than just validation of RL agents or quickly designing new environment settings. We now highlight these research questions for future work in Section 6.
>
> ### Relation to prior works using DSLs
> Thank you for pointing out two references that also use DSL languages, allowing us to better contextualize Griddly with prior work. We have updated Section 2.4 in the manuscript with these references, as well as a passage relating our work to these prior works.
>
> ### Does GriddlyJS only support TF.js agents?
> Thank you for bringing this to our attention. GriddlyJS supports any ONNX models (which can be converted to TF.js using common libraries). ONNX is a widely accepted format for neural network models and can be exported from most deep learning frameworks, e.g. PyTorch, JAX, TensorFlow. **We even include a script and instructions in the escape-rooms repository for easily converting Pytorch models to TF.js:** https://github.com/GriddlyAI/escape-rooms#using-checkpoints-in-griddlyjs. We have updated the manuscript to highlight these points in Appendix B.4.
>
> Lastly, we emphasize in Section 6 of the updated manuscript that we will be continuously improving the GriddlyJS interface and user experience as we gather more usage data and user feedback. Like many similar software tools, continuous maintenance and improvement is expected.
>
> We hope the clarifications above are sufficient for you to increase your support for our paper. Please let us know if you have any outstanding concerns that stand between us and a recommendation for acceptance.

---

> > ### Comment · Reviewer_pM8Q · 2022-08-23
> > **Updated Rating**
> >
> > I updated my rating and confidence level based on the response! Best wishes!!

---

### Meta-Review · Area_Chair_rQPg · 2022-09-05

**Recommendation:** Accept
**Confidence:** 5

**Metareview:**

The paper describes a web interface to design RL grid worlds based on the griddly engine.  This is a useful tool that lowers the barrier for experimenting with various RL grid worlds.  The authors addressed the main concerns of the reviewers who unanimously recommend acceptance.

---

### Decision · Program_Chairs · 2022-09-16

Accept